# Callovian Marine Reptiles of European Russia

**Nikolay Zverkov [1],\*, Maxim Arkhangelsky [2,3], Denis Gulyaev [4], Alexey Ippolitov [1,5,6] and Alexey Shmakov [7]**

[1]  Geological Institute of the Russian Academy of Sciences, Pyzhevsky Lane 7, Moscow 119017, Russia

[2]  Department of General Geology and Minerals, Saratov State University, Astrakhanskaya Str. 83, Saratov 410012, Russia

[3]  Department of Oil and Gas, Saratov State Technical University, Politekhnicheskaya Str. 77, Saratov 410054, Russia

[4]  Commission on Jurassic System of the Interdepartmental Stratigraphical Committee (ISC) of Russia, Chekhova St., 25/7, Yaroslavl 150054, Russia

[5]  School of Geography, Environment and Earth Sciences, Victoria University of Wellington Te Herenga Waka, 21 Kelburn Parade, Wellington 6012, New Zealand

[6]  Institute of Geology and Petroleum Technologies, Kazan Federal University, Kremlevskaya St., 18, Kazan 420008, Russia

[7]  Borissiak Paleontological Institute of the Russian Academy of Sciences, Profsoyuznaya St., 123, Moscow 117997, Russia

\*   Correspondence: zverkovnik@mail.ru

**Abstract:** Our knowledge of marine reptiles of the Callovian age (Middle Jurassic) is majorly based on the collections from the Oxford Clay Formation of England, which yielded a diverse marine reptile fauna of plesiosaurians, ichthyosaurians, and thalattosuchians. However, outside of Western Europe, marine reptile remains of this age are poorly known. Here, we survey marine reptiles from the Callovian stage of European Russia. The fossils collected over more than a century from 28 localities are largely represented by isolated bones and teeth, although partial skeletons are also known. In addition to the previously described rhomaleosaurid and metriorhynchids, we identify pliosaurids of the genera *Liopleurodon* and *Simolestes*; cryptoclidid plesiosaurians, including *Cryptoclidus eurymerus, Muraenosaurus* sp., and cf. *Tricleidus*, and ophthalmosaurid ichthyosaurians, including the iconic *Ophthalmosaurus icenicus*. These findings expand the ranges of several Callovian marine reptile taxa far to the Eastern Europe, and support the exchange of marine reptile faunas between Western and Eastern European seas in the middle to late Callovian. However, some specimens from the lower Callovian of European Russia show differences from typical representatives of the middle Callovian Oxford Clay fauna, possibly representing the earlier stages of evolution of some of these marine reptiles not yet recorded in Western Europe or elsewhere.

**Keywords:** Cryptoclididae; Ichthyosauria; Metriorhynchidae; Middle Jurassic; Middle Russian Sea; Plesiosauria; Pliosauridae; Paleobiogeography; Thalattosuchia

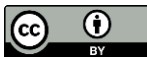

## 1. Introduction

The fossil record of marine reptiles is strongly spatially and temporally biased [1,2]. For the Jurassic period, the main sources of information are several formations in Western Europe, which have been the subject of intensive collecting and thus demonstrate "Lagerstätten effects" [1], along with a persisting global imbalance in the distribution of fossil data [3]. In the Middle Jurassic series, only its uppermost part, the Callovian, is well characterized by marine reptile fossils, and this knowledge is based nearly exclusively on the Oxford Clay Formation of England, which has yielded a diverse marine reptile fauna, including plesiosaurians, ichthyosaurians, and thalattosuchians [4–11]. Other Western European occurrences of Callovian marine reptiles in France and Germany yielded similar but scarcer faunas [12–26], whereas outside Western Europe, Callovian marine reptile occurrences are rarely reported. The records include localities in Argentina, Chile, Mexico

[27–30], Arctic Canada [31,32], and European Russia [33–40]. Specimens reported from these regions are mostly fragmentary and comprise isolated bones, teeth, and, more rarely, bone associations, all of which are identified in open nomenclature and generally referred to European taxa, with the only exception being a partial skeleton of a relict rhomaleosaurid, *Borealonectes russelli,* from Arctic Canada [31,32]. This situation hampers the assessment of the diversity of Callovian marine reptiles outside of Western Europe and demonstrates that our knowledge of Callovian marine reptile faunas is strongly spatially biased. The question arises, how justified can an extrapolation of the knowledge of the classic Oxford Clay fauna to a global scale be; whether marine reptile taxa were widespread in the Callovian, or whether there was some provincialism? In this respect, the growing number of discoveries of marine reptile remains from the Callovian of European Russia can be a valuable contribution to our knowledge. Here, we review all the available remains of plesiosaurians and ichthyosaurians from the Callovian of European Russia, taking into account both historical materials and recent finds. We also report new thalattosuchian finds, which supplement the recently described thalattosuchian specimens from Russia [40].

## 2. Historical Background

The earliest reports of marine reptile finds, probably from the Callovian Stage of European Russia, were made by Riabinin [41] and Bogolubov [42]. Riabinin described a pliosaurid bone association from the condensed upper Callovian and lowermost Oxfordian deposits of the Unzha River, near Gradulevo Village (currently part of Manturovo Town) in the Kostroma Region [41]. This specimen, now kept in the Mining Museum, St Petersburg (MM 219), was referred to *Peloneustes*; however, it was later re-considered as Pliosauridae indet. [43]. Bogolubov [42] described a new species, *Cryptoclidus simbirskensis*, from Gorodischi, Simbirsk Province (currently Ulyanovsk Region). This taxon was later considered Plesiosauria indet. [43,44]. The bones of "*Cryptoclidus simbirskensis"* were collected ex situ; therefore, their precise stratigraphic position is uncertain. Their state of preservation and color of the matrix occurs in the Callovian–lower Volgian interval of the region, although near Gorodischi the Callovian deposits are unknown. Bogolubov referred the specimen to the Callovian genus *Cryptoclidus,* based on its anteroposteriorly short cervical vertebrae with concave articular surfaces [42]. Therefore, the suggestion that this specimen is of Callovian age was the result of its taxonomic referral, not vice versa. Considering the presence of other cryptoclidids with similar morphology of cervical centra and propodials in stratigraphically younger deposits of the Arctic (i.e., *Djupedalia* [45] and an indeterminate cryptoclidid from Greenland [46]), we tentatively consider *Cryptoclidus simbirskensis* a nomen dubium, interpret its age as uncertain within the Callovian to lower Volgian interval, and exclude it from further consideration in our work.

In 1911, Bogolubov described several other plesiosaurian specimens [33], now kept in the V.I. Vernadsky State Geological Museum, Moscow (SGM). An elongated cervical vertebra (SGM 1358-37) from the Callovian of Alpatyevo, Ryazan Region, was identified as *Muraenosaurus* sp. Two articulated vertebrae from the lower Callovian of Gorky railway station (currently Fruktovaya), Ryazan Region [33] (pl. 1), were identified as "sacral vertebrae of *Pliosaurus* (*Liopleurodon*) sp." However, our examination of this specimen, SGM 1358-11, revealed that it is neither diagnostic nor complete enough to identify it more precisely than as the ventral portions of the pectoral or sacral centra of Plesiosauria indet., although their relatively large size implies an affinity with pliosaurids or rhomaleosaurids. A weathered caudal centrum (SGM 1358-53) from the middle Callovian of the Sysola River (Komi Republic) was referred to as "(?)*Cryptoclidus*" [33], but it is not possible to identify it more precisely than as Plesiosauria indet. An isolated tooth crown from the middle Callovian of Rechitsy, Moscow region, was described as a new species, *Thaumatosaurus calloviensis* [33]. This taxon was later referred to *Simolestes* as a valid species by Riabinin [47], synonymized with *Simolestes vorax* by Tarlo [6], and finally considered as Pliosauridae indet. by [43,44]. However, as argued by Tarlo [6] (p. 175), "*ridges which begin*

*part way up the crown on the external surface*" are a diagnostic feature of *Simolestes*, not yet found in any other pliosaurid genus (although, see "Morphotype 2" in [48]). Therefore, the Callovian tooth described and figured by Bogolubov ([33] pl. 2, figs. 1 and 6) can be referred to *Simolestes*. Unfortunately, this specimen is now lost.

In 1921, Milanovsky reported on the finding of a partial skeleton (SGM 1445-97−120, dorsal and caudal centra, partial pelvis and phalanges) of a large plesiosaur from the lower Callovian of the "Konnyi Barak" ravine, near the village of Verkhnaya Dobrinka, Zhirnovsk district, Volgograd Region [34]. This specimen was more thoroughly described and referred to as Rhomaleosauridae indet. by Benson et al. [36]

A finding of ichthyosaurian vertebra from the Callovian of Zamezhnaya, Komi Republic, was mentioned by Nesov et al. [49]. This specimen (ZIN PH 1/215) is depicted herein for the first time.

In 1999, Arkhangelsky described a partial forelimb (SSU 104a/27) from the (?)upper Callovian of Saratov Region [35]. He proposed a new genus and species *Khudiakovia calloviensis*, which was later considered synonymous with *Ophthalmosaurus icenicus* [50–52], or as a valid species of *Ophthalmosaurus* [53]. This specimen is reassessed herein.

More recent works on the Callovian marine reptiles from Russia include (in addition to the aforementioned contribution of Benson et al. [36]) a short report on a finding of the right maxilla of *Simolestes* from the lower Callovian of the Kostroma Region [37], a report on a finding of a partial ophthalmosaurid skeleton from the lower Callovian of the Republic of Mordovia [38], and an account of Jurassic marine reptiles from Moscow and adjacent territories [39] that provided brief descriptions and figures of pliosaurid and thalattosuchian tooth crowns, as well as several plesiosaurian and ichthyosaurian bones from the middle and upper Callovian of the Moscow and Ryazan regions. Finally, Young et al. [40] recently examined all the available thalattosuchian specimens from Russia and described, among others, several metriorhynchid tooth crowns from the Callovian.

## 3. Geological and Paleogeographic Settings

### 3.1. Paleogeographic Position of the Region

The Middle Jurassic epoch was a time of distribution of marine environments over the East European Platform. A transgression of shallow epicontinental seas far into its territory led to the formation of the Middle Russian Sea, which connected the Boreal seas and Tethys Ocean for a short time in the Early Bathonian [54–60]; then, after some regression during the mid-Bathonian, the Boreal–Tethys connection through this sea resumed in the course of a new cycle of transgression since the beginning of the Callovian [60–65]. Additionally, since the earliest Callovian, the Middle Russian Sea became connected to the seas of Central and Western Europe via the Dnieper-Donets basin and the sub-latitudinal Pripyat (Brest) Strait (Figure 1). This is evidenced by the exchange of the earliest Callovian cephalopod faunas between these regions [60,65–68]. This three-branch configuration of the Middle Russian Sea, with wide systems of straits connecting it to the northern, southern, and western seas and oceans, persisted to the early Volgian/Tithonian age of the Late Jurassic [69]. Thus, for circa 15 Myr, European Russia became a crossroad of migration routes between the Arctic (Boreal), Western European (Sub-Boreal and Sub-Mediterranean), and Western Tethys (tropical) sea basins. Besides, in the Callovian of the Middle Russian Sea, especially in its early chrons, an active neo-endemic evolution of cephalopods took place [65,70]. The eustatic sea-level fluctuations superimposed by non-uniform tectonic development of this vast territory resulted in complicated and changing coastal outlines of the Middle Russian Sea during its history and, consequently, complex stratigraphic architecture of the Jurassic System in European Russia [69].

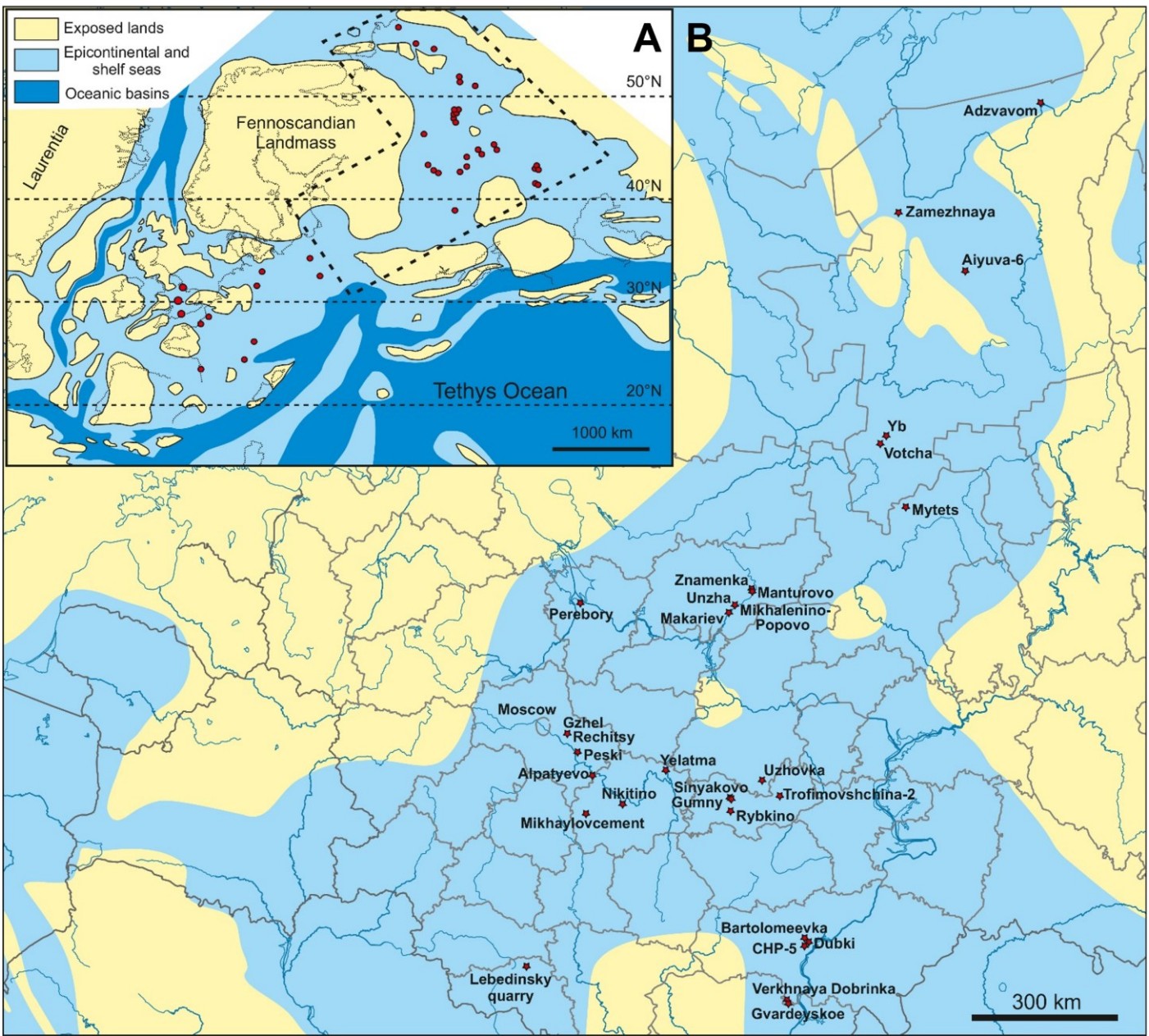

**Figure 1.** Localities of known marine reptile fossils from the Callovian of European Russia shown by red asterisks (**B**). Blue colorations on the map show outlines of the Middle Russian Sea during the Callovian, based on [71]. The upper left (**A**) shows Callovian paleogeography of Europe (after [72]; modified according to the data of [69,71]), with red dots showing the main localities of Callovian marine reptiles in Europe (i.e., numerous localities with the Oxford Clay Formation in England [4–11,73], localities in Northern and Southern France [12–21,74], Spain [75], Switzerland [76], Northern Germany [22–26], and Poland [77,78]). Dashed contour on (**A**) shows the area depicted in (**B**).

### 3.2. Localities with Marine Reptiles

At present, no less than 28 localities with Callovian marine reptile remains are known in European Russia (Figure 1). Some of these localities are historical and are not accessible for further excavation nowadays, and the knowledge of their geology is scarce, whereas others are still available or have been discovered and studied in detail in recent decades. We summarize the most important Callovian outcrops in Figures 2–4, based on published literature and personal observations of some of the authors (D.G., A.I., A.S., M.A.).



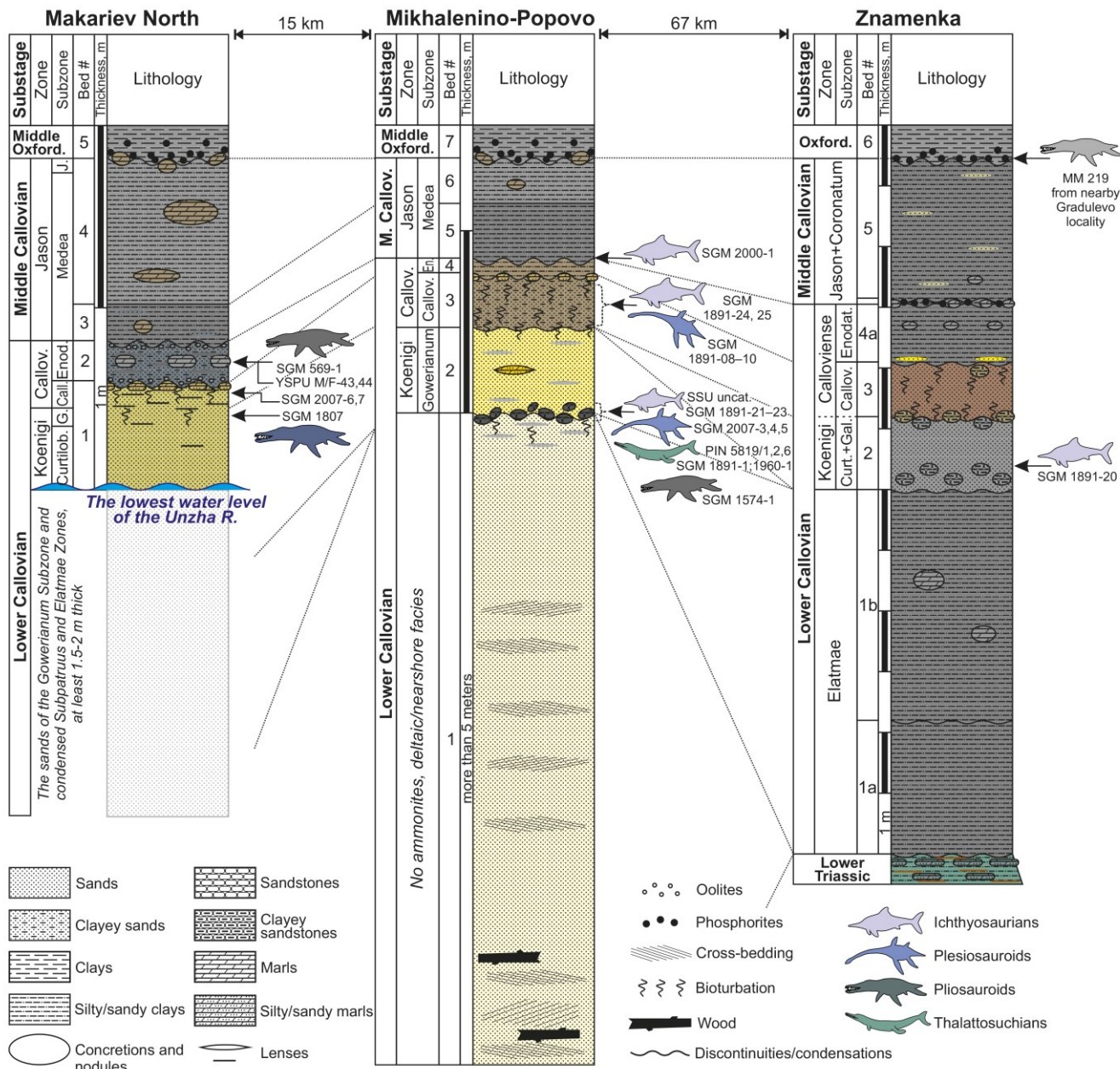

**Figure 2.** Stratigraphic sections of the Callovian marine reptile localities in the Unzha River basin (Kostroma Region) and distribution of marine reptile remains. Abbreviations of ammonite zones and subzones: Call/Callov, Calloviense; Curt/Curtilob, Curtilobum; En/Enod/Enodat, Enodatum; G/Gal, Galilaeii; J, Jason.

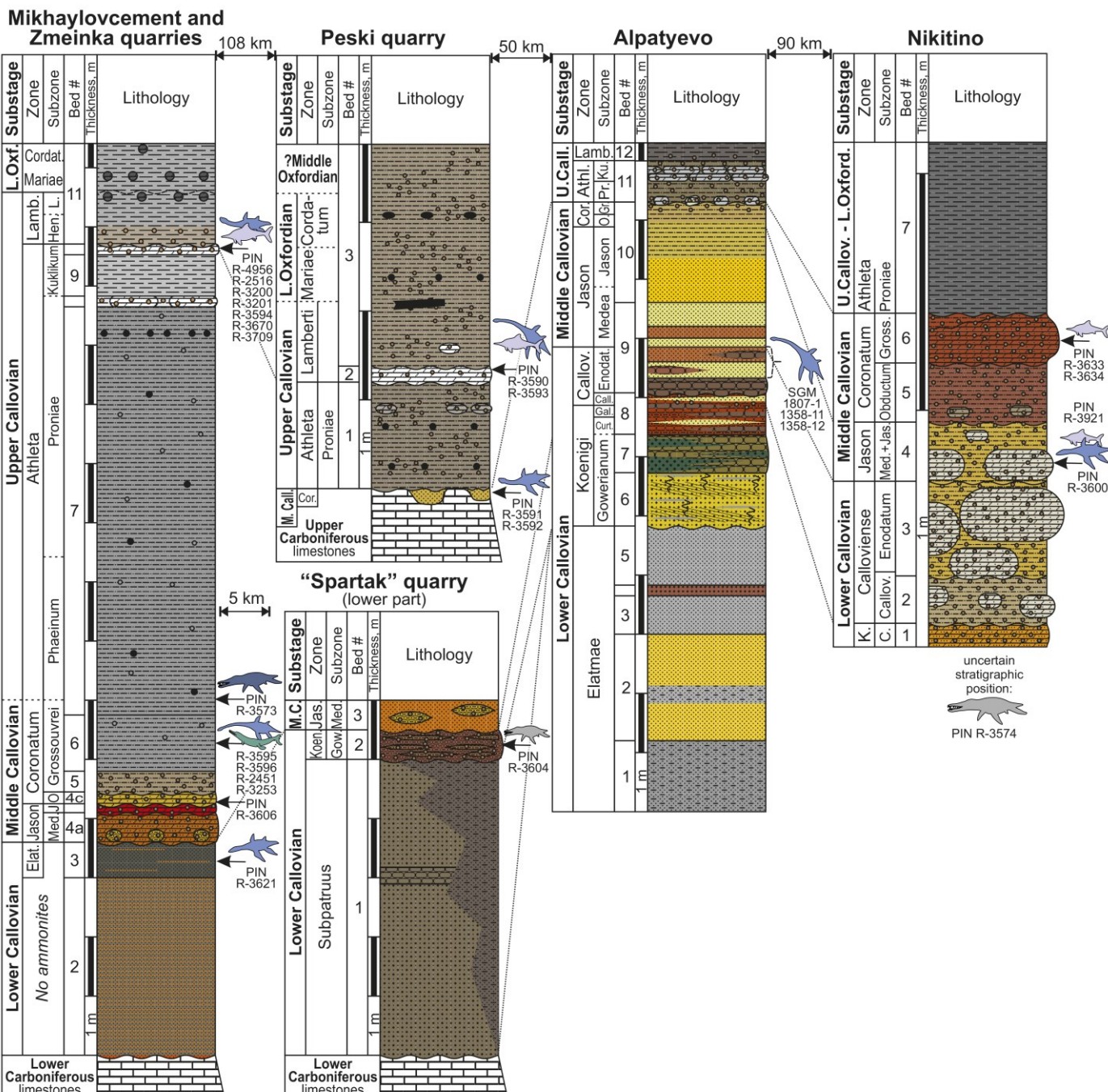

**Figure 3.** Stratigraphic sections of the Callovian marine reptile localities in Moscow and Ryazan regions and distribution of marine reptile remains. For lithology and other symbols of the legend see Figure 2. Abbreviations of ammonite zones and subzones: Athl, Athleta; C/Curt, Curtilobum; Call/Callov, Calloviense; Cor, Coronatum; Cordat, Cordatum; Enod/Enodat, Enodatum; Gal, Galilaeii; Gow, Gowerianum; Gr/Gross, Grossouvrei; Hen, Henrici; J/Jas, Jason; K/Koen, Koenigi; Ku, Kuklikum; L/Lamb, Lamberti; M/Med, Medea; O/Obd, Obductum; Ph, Phaeinum; Pr, Proniae. Mikhaylovcement section modified after [79]; Peski after [80]; Alpatyevo after [81].

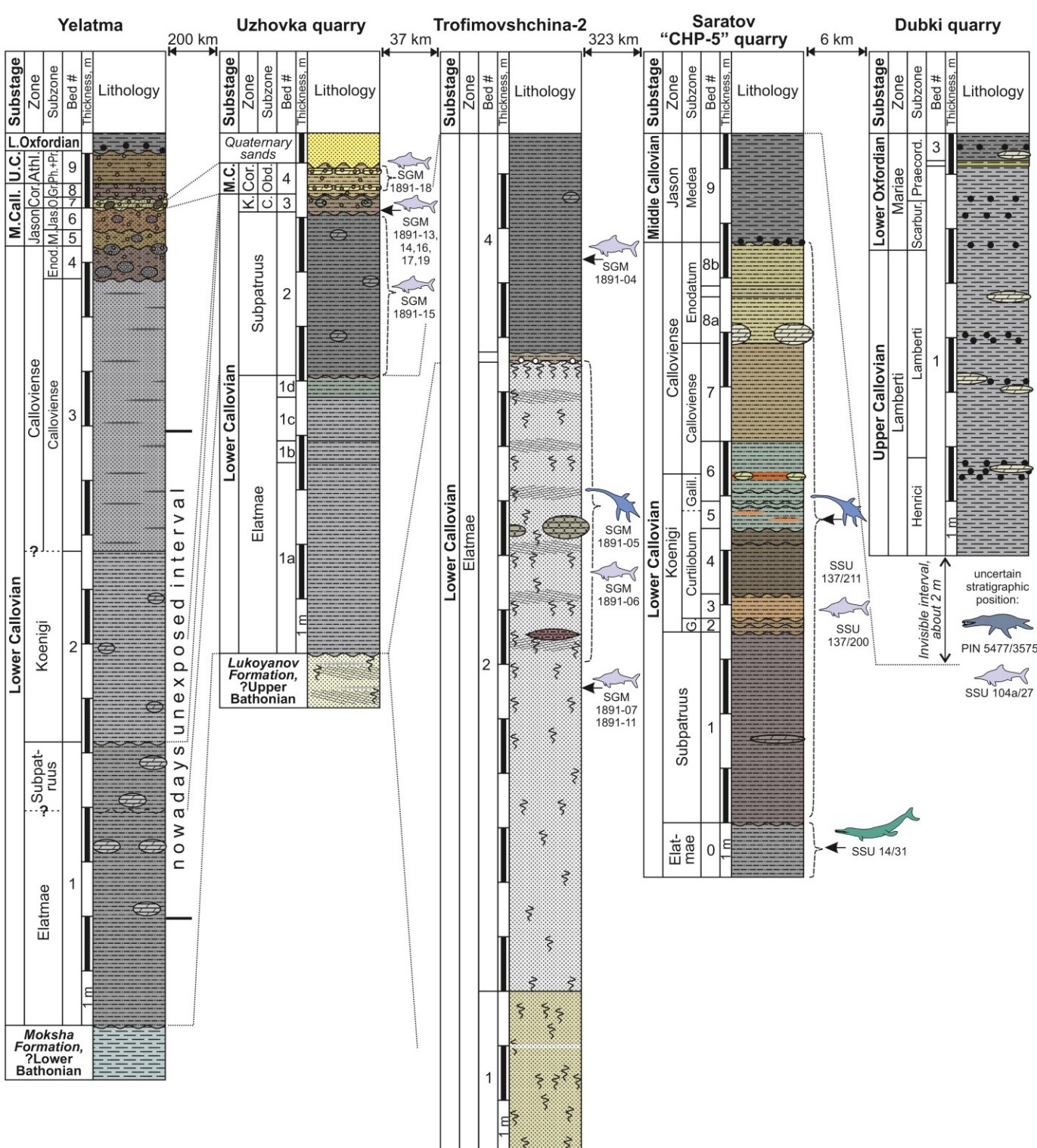

**Figure 4.** Stratigraphic sections of the Callovian marine reptile localities in Ryazan and Nizhny Novgorod regions, Republic of Mordovia, and Saratov Region. For lithology and other symbols of the legend, see Figure 2. Abbreviations of ammonite zones and subzones: Athl, Athleta; C, Curtilobum; Call/Callov, Calloviense; Cor, Coronatum; Enod, Enodatum; G, Gowerianum; Galil, Galilaeii; Gr–Grossouvrei; Jas, Jason; K, Koenigi; M, Medea; O/Obd, Obductum; Ph, Phaeinum; Pr, Proniae; Praecord, Praecordatum; Scarbur, Scarburgense. Yelatma locality section modified after [82–84]; Uzhovka and Trofimovshchina-2 originally drawn herein based on A.I.'s field descriptions; CHP-5 after [85]; Dubki after [86,87].

### 3.3. Geological Features of the Callovian of European Russia and Patterns of Stratigraphical Distributions of Marine Reptile Remains

The deposits of the Callovian age crop out in riverbanks, ravines, and quarries across the European part of Russia, commonly transgressively overlying the Late Paleozoic–Early Triassic and/or coastal–continental pre-Callovian Jurassic deposits in the western and eastern peripheries, and coastal-marine and marine pre-Callovian Jurassic in the central parts of European Russia [88,89]. Typically, bones and teeth of marine reptiles occur in condensed beds or above erosional surfaces, therefore, it might be difficult to date them to ammonite zone or even to the substage/stage level if they are found loose. The important feature derived from such a confinement is that, in most cases, vertebrate fossils are discovered only as disarticulated elements. Below, we list the most remarkable condensed intervals, which yielded the findings described in the present paper and have the highest potential for further discoveries of the Callovian marine reptiles.

At the north of the territory under consideration, in the Sysola River basin, there is a slight condensation at the boundary of the sandy Sysola Formation (Bathonian) and the clayey "Churkino" Formation (Callovian) [90–93], but the most prospective interval is the uppermost lower–middle Callovian, which is strongly condensed and packed with plenty of fossil invertebrates and vertebrates [90,91,93]. It is likely that most Callovian vertebrate fossils ever found in the region originate from this interval. Another example from the northern part of the region is shallow-water, strongly condensed sandy sections of the Callovian, located along the eastern shore of the paleobasin, adhering to the Ural Mountains (Adzvavom).

To the south, in the lower-middle reaches of the Unzha River Basin [84,94–96] (and D.G.'s unpublished data; see Figure 2), reptile remains can be associated with condensed sandy and sandy-clayey deposits of the lower Callovian (upper part of *Paracadoceras elatmae*, *Proplanulites koenigi,* and *Sigaloceras calloviense* zones), as well as to the disconformity between the middle Callovian and middle Oxfordian [41], and condensation sometimes covering the stratigraphic interval from the top of the *Kosmoceras jason* Zone to the bottom of the *Cardioceras densiplicatum* Zone.

Within the Callovian of the central part of European Russia (Figure 3), the condensed interval encompasses most of the lower Callovian and the middle Callovian in full. A very productive condensed interval is present at the transition from the *Cadochamoussetia subpatruus* Zone to the base of the *Pr. Koenigi* Zone. There is a well-defined erosional surface at the base of the middle Callovian (represented by the *K. jason* or *Erymnoceras coronatum* Zone), overlying different horizons of the lower Callovian. Finally, the remarkable erosional event occurred around the Callovian/Oxfordian boundary. In many places, middle and late Callovian strata are eroded to a significant depth, whereas the fauna of this age can be redeposited at the base of the lower/middle Oxfordian.

In the Middle Volga region [61,84,89,94,97–100] (Figure 4), reptile remains are often confined to the usually eroded and chronostratigraphically sliding boundary (upper Bathonian–lower Callovian) of the sandy Lukoyanov and clayey Yelatma formations. Thus, some of the finds originate from the upper Bathonian *Paracadoceras infimum* Zone, while some are from the lower part of the lower Callovian *P. elatmae* Zone. Furthermore, in the same region, well-defined condensed horizons are confined to the base of the lower Callovian *Pr. koenigi* Zone and to the base of the middle Callovian *E. coronatum* Zone. In the first case, the condensation interval may cover the top of the *P. elatmae* Zone, the entire *C. subpatruus* Zone, and the basal part of the *Pr. koenigi* Zone, while in the second, it expands from the top of the *P. elatmae* Zone to the base of the *E. coronatum* Zone. At the same time, the preserved deposits of the *Pr. koenigi–E. coronatum* zones are condensed and represented by oolitic facies [69]. In well-studied sections near Saratov [65,94], the main Callovian condensed levels fall on the boundary interval of the *C. subpatruus* and *Pr. koenigi* zones, and *S. calloviense /K. jason* zones, and the largest one, between the middle and upper Callovian, includes a biostratigraphic interval from the upper half of the *K. jason* Zone to the base of the *Lamberticeras lamberti* Zone.

In the western part of the former Middle Russian Sea, only the lower and middle Callovian are known [65,101–103]. In some sections (Mikhaylovsky Quarry, Kursk Region), there is a remarkable hiatus between the coarse sands of the subcontinental Arkino Formation (Bathonian?) and marine clayey deposits in the middle part of the *P. elatmae* Zone. Here, a lensing bed of rolled belemnite rostra is observed, among which fragments of hexacoral colonies occur (observed in 1997 by A.B. Guzhov, PIN RAS). In the same section, significant erosion occurs at the boundary of the *P. elatmae* and *C. subpatruus* zones. The middle Callovian is also present but not well-studied there [94]. The uppermost lower to middle Callovian interval contains multiple minor disconformities, each containing reworked fauna, and thus has a potential for marine reptile remains search.

## 4. Materials and Methods

More than a hundred of ichthyosaur, plesiosaur, and thalattosuchian specimens stored in 13 institutions are known from the Callovian of Russia. These are summarized in Table 1. The "Regional collection" of the Paleontological Institute RAS has a common catalogue number, 5477, which is abbreviated as "R" herein.

Prior to photography, teeth were coated with ammonium chloride to highlight the patterns of their enamel ornamentation.

The taxonomic frameworks used herein for reptilian taxa follow Benson and Druckenmiller [104] for plesiosaurians, Zverkov [105] for ichthyosaurians, and Johnson et al. [106] and Young et al. [107,108] for thalattosuchians. Ammonoid taxonomy and biostratigraphy follow Gulyaev and Ippolitov [65], and Kiselev [91].

Institutional abbreviations: **IG**, A.A. Chernov Geological Museum, Institute of Geology of the Komi Scientific Center of Ural Branch of RAS, Syktyvkar, Russia; **MM**, Mining Museum, St Petersburg, Russia; **MRUM**, Mordovian Republican United Museum of Local Lore, named after I.D. Voronin, Saransk, Russia; **NHMUK**, Natural History Museum, London, UK; **NNGASU**, Museum of Nizhny Novgorod State University of Architecture, Building and Civil Engineering, Nizhny Novgorod, Russia; **PIN**, Borissiak Paleontological Institute, Russian Academy of Sciences, Moscow, Russia; **PSM**, Penza State Museum of Local Lore, Penza, Russia; **SGM**, V.I. Vernadsky State Geological Museum of the Russian Academy of Sciences, Moscow, Russia; **SSU**, Saratov State University, Regional Museum of Earth Sciences, Saratov, Russia; **SSTU MEZ**, Museum of Natural History of Saratov State Technical University named after Y.A. Gagarin, Saratov, Russia; **TsNIGR**, F.N. Chernyshev Central Research Geological Survey Museum, St Petersburg, Russia; **UMLH**, Ukhta Museum of Local History, Ukhta, Russia; **YSPU**, Museum at K.D. Ushinsky Yaroslavl State Pedagogical University, Yaroslavl, Russia; **ZIN PH**, Paleoherpetological Collection, Zoological Institute of the Russian Academy of Sciences, St. Petersburg, Russia.

**Table 1.** Catalogue of marine reptile remains from the Callovian of European Russia.

| Repository and Catalogue No, Status | Identification (Historical in Parentheses) | Material | Locality | Stratigraphic Position | Reference |
|---|---|---|---|---|---|
| ZIN PH 1/215 | Ichthyosauria indet. | posterior dorsal centrum | Zamezhnaya, "Outcrop 14" (label info; but "outcrop 16" in [109]), Komi Republic | middle/upper Callovian boundary strata, beds with *Longaeviceras* | collected by V.S. Kravets |
| Specimen location unknown | Ichthyosauria indet. | centrum | Adzvavom, Inta District, Komi Republic, "outcrop 14"in [110] | Upper Callovian, *Longaeviceras nikitini* Zone (Boreal standard) | [110] |

| | | | | | |
|---|---|---|---|---|---|
| UMLH MPZ KP 1982/1 | Ichthyosauria indet. | anterior presacral centrum | Aiyuva-6, Sosnogorsk District, Komi Republic | middle Callovian? | [111] |
| IG 722/6 | Ichthyosauria indet. | posterior caudal centrum | Aiyuva-6 | middle Callovian, *Rondiceras milaschevici* Zone (Boreal standard) | [111] |
| IG 93/13 | Ichthyosauria indet. | anterior presacral centrum | Votcha, Sysola District, Komi Republic | ex situ, likely from the middle Callovian, *K. jason–E. coronatum* zones | [112] |
| SGM 1358-53 | Plesiosauria indet. ("*Cryptoclidus* (?) sp." in [33]) | caudal vertebra | Kargort (Yb) at Sysola River, Komi Republic | middle Callovian | [33] (p. 230) |
| B.A. Mal'kov's private collection | Plesiosauria indet., Pliosauridae indet., Ichthyosauria indet., *Muraenosaurus* sp. | cervical and dorsal vertebrae, plesiosaurian distal propodial | Kargort (Yb) | middle Callovian? | [112,113] |
| SGM uncatalogued | Ichthyosauria indet. | cast of posterior dorsal centrum | Mytets River, Sychugovy Village (abandoned), Nagorsk District, Kirov Region | Callovian? | [114] collected by Khabakov, 1924 |
| NNGASU 147/2080 | Pliosauridae indet. | proximal portion of a large propodial bone | Vyatka–Kama phosphorite field, Verkhnekamsky District, Kirov Region | middle Jurassic, Callovian? | collected by Yu. S. Tamoykin |
| MM 219 | Pliosauridae indet. | partial postcranial skeleton | Gradulevo, Manturovo District, Kostroma Region | middle Callovian or middle Oxfordian | [41] |
| SGM 1891-02, 03 and 20 | cf. *Ophthalmosaurus* | anterior dorsal vertebra, indet. bone fragments, and partial mandible with teeth. | Znamenka, Manturovo District, Kostroma Region | lower Callovian, *Pr. koenigi* Zone, *G. galilaeii* Subzone | collected by V.V. Mitta, 2015 |
| SGM 1807 | *Liopleurodon ferox* | isolated posterior tooth and partial postcranial skeleton; small plesiosaurian vertebra found in association | Makariev North locality, Makariev District, Kostroma Region | lower Callovian, upper part of *Pr. koenigi* Zone, *G. galilaeii* Subzone | collected by A.V. Stupachenko |
| SGM 2007-6,7 | *Liopleurodon* sp. | three associated teeth | Makariev, Makariev District, Kostroma Region | lower Callovian, *S. calloviense* Zone | collected by A.V. Stupachenko, 2016 |
| SGM 569-1 | *Simolestes* sp. | propodial, likely femur | Makariev | lower Callovian, *S. calloviense* Zone, *C. enodatum* Subzone | collected by A.V. Stupachenko and V.V. Mitta |
| YSPU M/F-45 | Pliosauridae indet. | articular surface of a large vertebral centrum | Makariev North | lower Callovian, *S. calloviense* Zone | collected by D.N. Kiselev |

| YSPU M/F-44 | cf. *Cryptoclidus* | nearly complete dorsal vertebra | Makariev North | same as above | collected by D.N. Kiselev |
|---|---|---|---|---|---|
| SGM 1574-1 | *Simolestes* sp. | left maxilla | Mikhalenino, Makariev District, Kostroma Region | lower Callovian, horizon of reworked fauna of *P. elatmae* and *C. subpatruus* zones at the base of *G. gowerianum* Sub-zone | [37] collected by A.V. Stupachenko |
| SGM 1891-10 | cf. *Muraenosaurus* | posterior cervical vertebra | Mikhalenino | lower Callovian, *S. calloviense* Zone | collected by A.V. Stupachenko, 2001 |
| SGM 1891-09 | cf. *Muraenosaurus* | posterior cervical vertebra | Mikhalenino | same as above | collected by A.V. Stupachenko |
| SGM 1891-08 | *Muraenosaurus* sp. | anterior cervical vertebra | Mikhalenino | same as above | collected by A.V. Stupachenko, |
| SGM 2007-3(1–5) | *Muraenosaurus* sp. | five isolated teeth | Mikhalenino | lower Callovian, horizon of reworked fauna of *P. elatmae* and *C. subpatruus* zones at the base of *G. gowerianum* Sub-zone | collected by A.V. Stupachenko, 1999 |
| PIN 5819/3 | *Muraenosaurus sp.* | partial tooth | Mikhalenino | same as above | collected by A.V. Stupachenko, 1999 |
| SGM 2007-4 | cf. *Tricleidus* | tooth crown | Mikhalenino | same as above | collected by A.V. Stupachenko, 1999 |
| SGM 2007-5 | Pliosauridae indet. | fragmant of a large tooth | Mikhalenino | same as above | collected by A.V. Stupachenko, 2016 |
| SGM 2007-8 | cf. *Ophthalmosaurus* | tooth | Mikhalenino | same as above | collected by A.V. Stupachenko, |
| PIN 5819/4 | cf. *Ophthalmosaurus* | tooth crown | Mikhalenino | same as above | collected by A.V. Stupachenko, 1999 |
| SGM 1891-22 | cf. *Ophthalmosaurus* | left jugal | Mikhalenino | same as above | collected by A.V. Stupachenko, 1999 |
| SSU uncatalogued | cf. *Ophthalmosaurus* | left quadrate | Mikhalenino | same as above | collected by A.V. Stupachenko, 2001 |
| SSU uncatalogued | Ophthalmosauria indet. | right nasal, antero-dorsal centrum, rib | Mikhalenino | same as above | collected by A.V. Stupachenko, 1998 |
| SGM 1891-21 | Ophthalmosauria indet. | a series of nine tail fluke centra with neural arches | Mikhalenino | same as above | collected by A.V. Stupachenko, 1999 |

| | | | | | |
|---|---|---|---|---|---|
| SGM 1891-23 | Ichthyosauria indet. | long dorsal rib (50 cm proximoditally) | Mikhalenino | same as above | collected by A.V. Stupachenko, 1999 |
| SGM 1960-01 | Geosaurini indet. | tooth crown | Mikhalenino | same as above | [40] (as "SGM BX-12") collected by A.V. Stupachenko |
| SGM 1891-01 | Thalattosuchia indet. | autopodial element | Mikhalenino | same as above | Collected by A.V. Stupachenko |
| PIN 5819/1, 2, 6 | Metriorhynchidae indet. | three miscellaneous teeth | Mikhalenino | same as above | [40] |
| SGM 1891-25 | Ichthyosauria indet. | posterior presacral centrum | Mikhalenino | lower Callovian, *S. calloviense* Zone | collected by A.V. Stupachenko, |
| SGM 1891-24 | Ichthyosauria indet. | caudal centrum | Mikhalenino | same as above | collected by A.V. Stupachenko, |
| SGM 2000-1 | *Ophthalmosaurus icenicus* | left premaxilla | Mikhalenino | clays at the boundary of the lower and middle Callovian | collected by A.V. Stupachenko, 1999 |
| SGM 1961 | *Ophthalmosaurus icenicus* | skull fragments including quadrate, angular and tooth; atlas-axis and vertebrae from all regions of the column; fragments of pectoral girdle and partial forelimb | Perebory, Rybinsk District, Yaroslavl Region | middle Callovian; *K. jason* Zone, *K. medea* Subzone | collected by K.K. Kotov and N.Z., 2021. |
| specimen is lost, holotype | *Simolestes* sp. (*Thaumatosaurus calloviensis*) | isolated tooth crown | Rechitsy near Gzhel village, Ramenskoe District, Moscow Region | middle Callovian | [33] (p. 200, pl. II, figs. 1 and 6) |
| PIN R-3589 | cf. *Simolestes* | partial crown | Gzhel, Ramenskoe District, Moscow Region | middle Callovian | collected by M.S. Boiko, 1990 |
| PIN 5818/8 | Pliosauridae indet. | partial crown | Rechitsy | middle Callovian | collected by P.A. Gerasimov, 1928 |
| PIN 5818/9 | *Tyrannoneustes* sp. | tooth crown | Rechitsy | middle Callovian, likely *E. coronatum* Zone | [40] collected by P.A. Gerasimov, 1928 |
| PIN 5819/7 | Metriorhynchidae indet. | tooth crown | Rechitsy | middle to lower upper Callovian | [40] |
| PIN R-3590 | cf. *Muraenosaurus* | posterior cervical centrum | Peski Quarry, Kolomensk District, Moscow Region | upper Callovian, *L. lamberti* Zone | [39] (fig. 5b) collected by A. Kuzmenko, 2015 |
| PIN R-3591, PIN R-3592 | Cryptoclididae indet. | cervical of juvenile | Peski Quarry | middle Callovian, *E. coronatum* Zone | [39] (fig. 5d); collected by N. Denisova and M. Sushko, 2015 |
| PIN R-3593 | Ichthyosauria indet. | four posterior caudal centra | Peski Quarry | upper Callovian, *L. lamberti* Zone | collected by I.V. Ilyasov, 1990s |

| | | | | | |
|---|---|---|---|---|---|
| SGM 1358-11 | Plesiosauria indet. | two articulated pectoral or sacral centra | Gorky (nowadays Fruktovaya Station), Lukhovitsy District, Moscow Region | lower Callovian, likely *S. calloviense* Zone, *C. enodatum* Subzone | [33] (p. 149, pl. I) |
| SGM 1358-12 | Plesiosauria indet. | proximal fragment of propodial | Alpatyevo, Lukhovitsy District, Moscow Region | lower Callovian? | [33] (p. 151) |
| SGM 1358-37 | *Muraenosaurus* cf. *leedsi* | cervical vertebra | Alpatyevo | lower Callovian, likely *S. calloviense* Zone, *C. enodatum* Subzone | [33] (p. 235, pl. III, figs. 1 and 2) |
| PIN R-3600 | *Cryptoclidus eurymerus* | partial skeleton, including several cervical, pectoral, dorsal and caudal vertebrae, dorsal ribs and gastralia, partial pubis, humerus, radii, femora | Nikitino, Spasskiy District, Ryazan Region | middle Callovian, *K. jason* Zone | collected by the Club of Junior Paleontologists of the Paleontological Museum, lead by A.S., in 2014–2015 |
| PIN R-3574 | Pliosauridae indet. | partial tooth crown | Nikitino | middle Callovian | collected by D.N. Kasantsev, 2011 |
| PIN R-3633, 3634 | Ichthyosauria indet. | caudal centrum and jawbone fragments | Nikitino | middle Callovian, *E. coronatum* Zone, *K. grossouvrei* Subzone | collected by the Club of Junior Paleontologists of the Paleontological Museum, lead by A.S., in 2014-2015 |
| PIN R-3573 | *Liopleurodon ferox* | large caniniform tooth crown | Mikhaylovcement Quarry, Mikhaylov District, Ryazan Region | same as above | [39] collected by A. Kuraev and A.S., 2012 |
| V. Bakhtin private collection | *Simolestes* sp. | partial tooth crown | Zmeinka Quarry, Mikhaylov District, Ryazan Region | upper Callovian | [39] (fig. 1C) |
| PIN R-3604 | Pliosauridae indet. | tooth crown fragment | Spartak Quarry, Mikhaylov District, Ryazan Region | lower Callovian, *Pr. koenigi* Zone | collected by A.S. |
| PIN R-3595 | cf. *Muraenosaurus* | posterior cervical vertebra | Mikhaylovcement Quarry | middle Callovian, *E. coronatum* Zone, *K. grossouvrei* Subzone | [39] (fig. 5c) collected by A. Churkin |
| PIN R-3621 | *Cryptoclidus* sp. | tooth crown | Mikhaylovcement Quarry | lower Callovian, upper part of *P. elatmae* Zone | collected by D.G., 2017 |
| PIN R-3670 | Cryptoclididae indet. | cervical vertebra | Mikhaylovcement Quarry | upper Callovian, *P. athleta* Zone, *K. kuklikum* Subzone | collected by S.V. Grishin, 2017 |

| PIN R-3709 | Cryptoclididae indet. | pectoral vertebra | Mikhaylovcement Quarry | same as above | collected by Ershova O.G. |
|---|---|---|---|---|---|
| PIN R-3606 | Cryptoclididae indet. | vertebra | Mikhaylovcement Quarry | middle Callovian, *E. coronatum* Zone, *K. obductum* Subzone | collected by Ershova O.G. |
| PIN R-3201, PIN R-3202 | Plesiosauria indet. | caudal centrum and dorsal neural arch fragment | Mikhaylovcement Quarry | upper Callovian, *P. athleta* Zone, *K. kuklikum* Subzone | collected by K. Nazarov, 2011 |
| PIN R-3596 | Plesiosauia indet. | posterior caudal centrum | Mikhaylovcement Quarry | middle Callovian, *E. coronatum* Zone, *K. grossouvrei* Subzone | collected by S. Rossiysky, 2016 |
| PIN R-4956 | *Ophthalmosaurus icenicus* | humerus, radius, intermedium, radiale | Zmeinka Quarry | upper Callovian, *P. athleta* Zone *K. kuklikum* Subzone | collected by L.V. Kulagina, 2023 |
| PIN R-2516 | cf. *Ophthalmosaurus* | anterior accessory epipodial element | Mikhaylovcement Quarry | same as above | [39] (fig. 12D) collected by A.S., 2010 |
| PIN R-3200 | Ichthyosauria indet. | posterodorsal centrum | Mikhaylovcement Quarry | same as above | collected by N. Ushakov, 2011 |
| PIN R-3594 | Ichthyosauria indet. | dorsal centrum | Zmeinka Quarry | upper Callovian, *L. lamberti* Zone | collected by I.A. Dadykin, 2016 |
| PIN R-2451 | *Tyrannoneustes* sp. | tooth crown | Mikhaylovcement Quarry | middle Callovian, *E. coronatum* Zone, *K. grossouvrei* Subzone | [39,40] collected by A.S., 2010 |
| PIN R-3253 | cf. *Thalattosuchus* | tooth crown | Mikhaylovcement Quarry | same as above | [39,40] collected by A.S. Shmakov, 2012 |
| SGM w/o number | Geosaurinae indet. | cervical vertebra | Mikhaylovcement Quarry | middle to upper Callovian; collected ex situ | collected by K. Volkov |
| TsNIGR 157a/649 | Pliosauridae indet. | sacral vertebra | Yelatma, Kasimov District, Ryazan Region | indet. Callovian | collected by N.A Bogoslovsky |
| TsNIGR 144/1712 | cf. *Muraenosaurus* | dorsal vertebra | Yelatma | indet. Callovian | collected by E.M. Lutkevich, 1925 |
| SGM 1891-13, 14, 16, 17, 19 | Ichthyosauria indet. | weathered centrum (17), anterior to middle caudal centrum (19); posterior caudal centrum (14); coracoid lateral fragment (13); small rib fragment (16) | Uzhovka Quarry, Pochinki District, Nizhny Novgorod Region | lower Callovian, lower part of *Pr. koenigi* Zone | collected by D.G., 1995 |
| SGM 1891-18 | Ichthyosauria indet. | small caudal vertebra | Uzhovka Quarry | middle Callovian, *K. jason* Zone, *K. obductum* Subzone | collected by D.G., 1995 |
| SGM 1891-15 | Ichthyosauria indet. | posterior dorsal centrum | Uzhovka Quarry | lower Callovian, *P. elatmae or C. subpatruus* zones | collected by D.G., 1995 |

| | | | | | |
|---|---|---|---|---|---|
| SGM 1891-05 | Plesiosauria indet. | dorsal centrum of juvenile | Trofimovshchina-2, Romodanovo District, Republic of Mordovia | lower Callovian, *P. elatmae* Zone | collected by N.Z., 2018 |
| SGM 1891-06 | *Ophthalmosaurus* cf. *calloviensis* | radius | Trofimovshchina-2 | same as above | collected by N.Z., 2018 |
| SGM 1891-04 | Ichthyosauria indet. | middle dorsal centrum | Trofimovshchina-2 | lower Callovian, *P. elatmae* Zone, upper clayey part | collected by A.P. Ippolitov, 2016 |
| SGM 1891-07 | Ichthyosauria indet. | rib | Trofimovshchina-2 | same as above | collected by A. I., 2016 |
| SGM 1891-11 | Ichthyosauria indet. | weathered vertebral centrum and proximal rib fragments | Trofimovshchina-2 | same as above | collected by A. I. and N.Z., 2018 |
| PSM 3999-4004 | *Ophthalmosaurus* cf. *calloviensis* | jaw fragments with teeth; sclerotic ring; dorsal and caudal centra; neural arches; proximal fragment of humerus; radius, ulna and phalanges | Moksha River near Rybkino Village, Kovylkino District, Republic of Mordovia | lower Callovian, *P. elatmae* to *Pr. koenigi* zones | collected by A.A. Stuckenberg, 1925 |
| Collection of V.M. Efimov | *Ophthalmosaurus* sp. | anterior part of the skeleton including skull, ribs and limb elements | Sinyakovo Village, Krasnoslobodsk District, Republic of Mordovia | lower Callovian, *P. elatmae* Zone | [38] |
| MRUM 1315/1 | cf. *Thalattosuchus* | tooth crown | Gumny, Krasnoslobodsk District, Republic of Mordovia | lower Callovian, *P. elatmae* to *Pr. koenigi* Zones | [40] |
| SGM 1891-27 | Plesiosauria indet. | scapular fragment | Lebedinsky Quarry; Belgorod Region | lower Callovian, *P. elatmae* Zone | collected by N.Y. Bragin, 2000 |
| SSU 137/211 | cf. *Muraenosaurus* | cervical vertebra | CHP-5 (TETs-5), Saratov, Saratov Region | lower Callovian, *C. subpatruus–S. calloviense* zones | collected 1985 |
| SSU 137/200 | Ichthyosauria indet. | caudal centrum | sovkhoz Leninskiy Put' near CHP 5 power station (TETs-5), Saratov, Saratov Region | lower Callovian, *P. elatmae* Zone | collected by M.A., 1984 |
| SSU 14/31 (historical No 104a/29) | Geosaurini indet. | tooth crown | sovkhoz Leninskiy Put' near CHP 5 | lower Callovian, *P. elatmae* Zone | [35,40] collected by M.A., 1984 |
| SSU 104a/27, holotype | *Ophthalmosaurus calloviensis* (*Khudiakovia calloviensis*) | partial forelimb | construction pit near Dubki Village, Saratov District | upper Callovian? (or lower Callovian) | [35] collected 1977 by the Children's Club of Local History of Saratov |
| PIN R-3575 | Pliosauridae indet. | Crown fragment | Dubki, Saratov District, Saratov Region | upper Callovian, *L. lamberti* Zone | collected by R. Yu. Stredinin, 2011 |

| SGM 1445-97–120 | Rhomaleosauridae indet. | Ilium, partial ischium and pubis; dorsal and caudal centra; rib, metatarsals and phalanges | Verkhnaya Dobrinka, Zhirnovsk district, Volgograd Region | lower Callovian, *P. elatmae* Zone | [34,36] collected by E.V. Milanovsky, 1920 |
| SSTU MEZ 3/4 | cf. *Ophthalmosaurus* | partial skeleton | Gvardeyskoe Village, Krasnoarmeysk District, Saratov Region | lower Callovian, *P. elatmae* Zone | donated to SSTU by S.A. Bratashova in early 2000s |
| Whereabouts of specimen unknown | Ichthyosauria indet. | Partial skeleton | 0.5 km south of Bartolomeevka, Saratov District, Saratov Region | lower Callovian, *P. elatmae* Zone | [115] collected by V. Grizbovskiy, 1998 |

## 5. Results of the Study of the Marine Reptile Material

### 5.1. Pliosaurids

The majority of marine reptile remains collected from the Callovian deposits of European Russia are plesiosaurian teeth and vertebrae. Among these, robust teeth of pliosaurids are quite common.

#### 5.1.1. *Liopleurodon ferox*

The specimen, PIN R-3573, is a very large caniniform tooth crown with a partial root. The apicobasal length of the crown is 123 mm, and its basal diameter is 51 mm. This tooth crown is the largest ever reported for Callovian pliosaurids, among which no specimens with crowns exceeding 110 mm in apicobasal height were hitherto reported [85,116]. PIN R-3573 can be referred to the genus *Liopleurodon*, as it is consistent with the holotypic crown in its large size, circular cross-section, coarse enamel ridges triangular in cross-section, rare on the labial surface, numerous on the lingual surface, and all reaching the base of the crown [12,116]. The crown of PIN R-3573 is slightly curved and terminates in a pointed, sharp apex. The labial (outer convex) side bears only two small ridges in its basal part (Figure 5A); the rest of its surface is subtly rugose, with numerous small ridglets. The lingual, mesial, and distal surfaces bear numerous sharp and nearly straight ridges, which emerge at the base of the crown and extend towards the apex to a varying extent. Only five of these reach the tip (Figure 5D). The ridges enlarge towards the apex, being approximately 0.5 mm wide at the base of the crown and more than doubling in width towards the apex.

In addition to PIN R-3573, the crown morphology described above is present in three associated teeth, SGM 2007-6,7 (Figure 5J–M), from the lower Callovian of the Unzha River. Furthermore, one tooth with such morphology (Figure 5E–I) was found in association with the postcranial skeleton SGM 1807, which is described below. The largest of these teeth (SGM 2007-7) is similar to the above-described PIN R-3573, whereas smaller teeth are posterior "ratchet"-type teeth with stout and strongly curved crowns bearing frequent ridges around their entire circumference (Figure 5E–L).

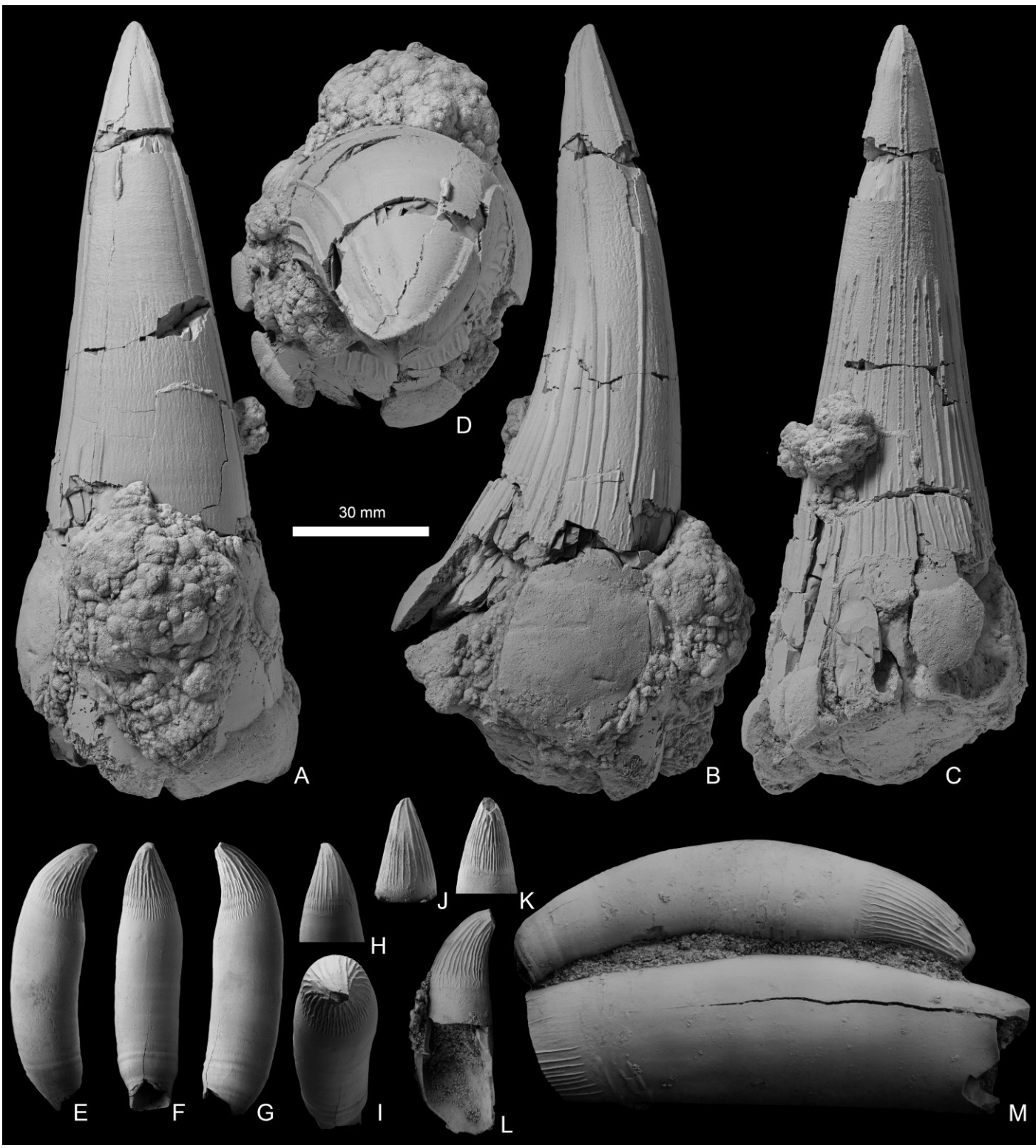

**Figure 5.** Teeth of *Liopleurodon ferox* from the Callovian of European Russia. Mesial ("caniniform" type) teeth PIN R-3573 (**A–D**) from the middle Callovian of Ryazan Region, and SGM 2007-7 (**M**) from the lower Callovian of Makariev, Kostroma Region. Distal ("ratchet" type) teeth SGM 1807 (**E–I**) and SGM 2007-6 (**J–L**) from the lower Callovian of Makariev, Kostroma Region. Views: anterior or posterior (**D,E,G,L**), lingual (**C,F,K**), labial (**B,H,J**), and apical (**D,I**).

A partial skeleton of a large pliosaurid, SGM 1807, unearthed by Andrey V. Stupachenko in the Makariev North locality at the Unzha River, Kostroma Region, is among the most significant marine reptile discoveries in the Callovian of European Russia to date. This specimen can be referred to the genus *Liopleurodon* based on the following features: a tooth crown that is conical in shape, bearing well-spaced and coarse ridges, all reaching the crown base, and the femur with elongated diaphysis giving the bone *"almost parallel sides for the essential part of its length"*, as diagnosed by Tarlo [6] (p. 167). No cranial remains were found for SGM 1807, except a single tooth (Figure 5E–I). The total length of the tooth, including the crown and root, is 56 mm. The crown apicobasal height is 18 mm, and the basal diameter is 13 mm.

The vertebrae are not preserved in SGM 1807; however, the cervical, dorsal, and sacral ribs are present. Anterior cervical ribs are robust, double-headed, with widely spaced capitulum and tuberculum (Figure 6P–V). Their anterolateral edge is tapered, their posterior surface bears a pronounced trough (Figure 6T,U), and their distal end is expanded, forming anterior and posterior processes, with the posterior process being more prominent than the anterior (Figure 6P–S). Five complete dorsal ribs and multiple rib fragments are preserved. The dorsal ribs are large and massive (the longest preserved rib is 550 mm long and 45 mm in cross-section diameter). The dorsal ribs are curved and circular in cross-section (Figure 6W–Y). One sacral rib is preserved; it is short and robust, possessing anteroposteriorly expanded proximal and distal ends, which are twisted relative to one another (Figure 6Z).

Both coracoids are partially preserved, lacking posteromedial portions of the plates. They are large and thin elements with a mediolateral width of 450 mm. The thickest part of the bone is the glenoid region. The angle between the glenoid and scapular surfaces is obtuse and measures about 150°. The glenoid contribution is strongly inclined anteriorly, with its posterior portion markedly protruding laterally, unlike in any other Callovian pliosaurid, except for *Liopleurodon* (N.Z.'s pers. obs. on NHMUK PV R2738; see text-fig. 5 in [5]). This is an additional feature supporting our generic referral of SGM 1807. The scapular facet is triangular in outline and gradually transitions medially into the anterior coracoid shelf (Figure 6B,D). The medial articular facet of the coracoid is S-curved with convex dorsal and concave ventral surfaces in the thickest part (Figure 6C). The anteromedial process is thus directed anteroventrally. A bilateral element of problematic identity may represent an interclavicle. It consists of two slender and compressed rami and a short posterior process (Figure 6K,L). It is 40 cm wide. If such identification is correct, SGM 1807 is among the very few post-Liassic pliosaurids known to have a clavicle and/or interclavicle (clavicles and/or interclavicles are reported for *Peloneustes* [5,10], *Marmornectes* [9], and *Eardasaurus* [11]).

The pelvic girdle is preserved as two fragmental pubes and an almost complete left ischium. The ischium measures 750 mm anteroposteriorly, from its posterior end to the anterior surface of the acetabular facet. It is 445 mm wide mediolaterally, from the acetabulum to the medial symphysis, and therefore has a length-to-width ratio of 1.69. The medial and lateral surfaces of the ischiadic blade are approximately parallel posteriorly, converging only weakly. The ischial neck, connecting the acetabular process to the ischial blade, is relatively wide due to shallow excavations of anterior and posterior edges, and its anterior border is sharply edged. The acetabular process of the ischium is 230 mm long anteroposteriorly and 110 mm high dorsoventrally. It is divided into facets for the ilium (facing posterolaterally), acetabulum (facing laterally), and pubis (facing anteriorly). Overall, this element is in agreement with the ischia of *Liopleurodon* from the Oxford Clay [5].

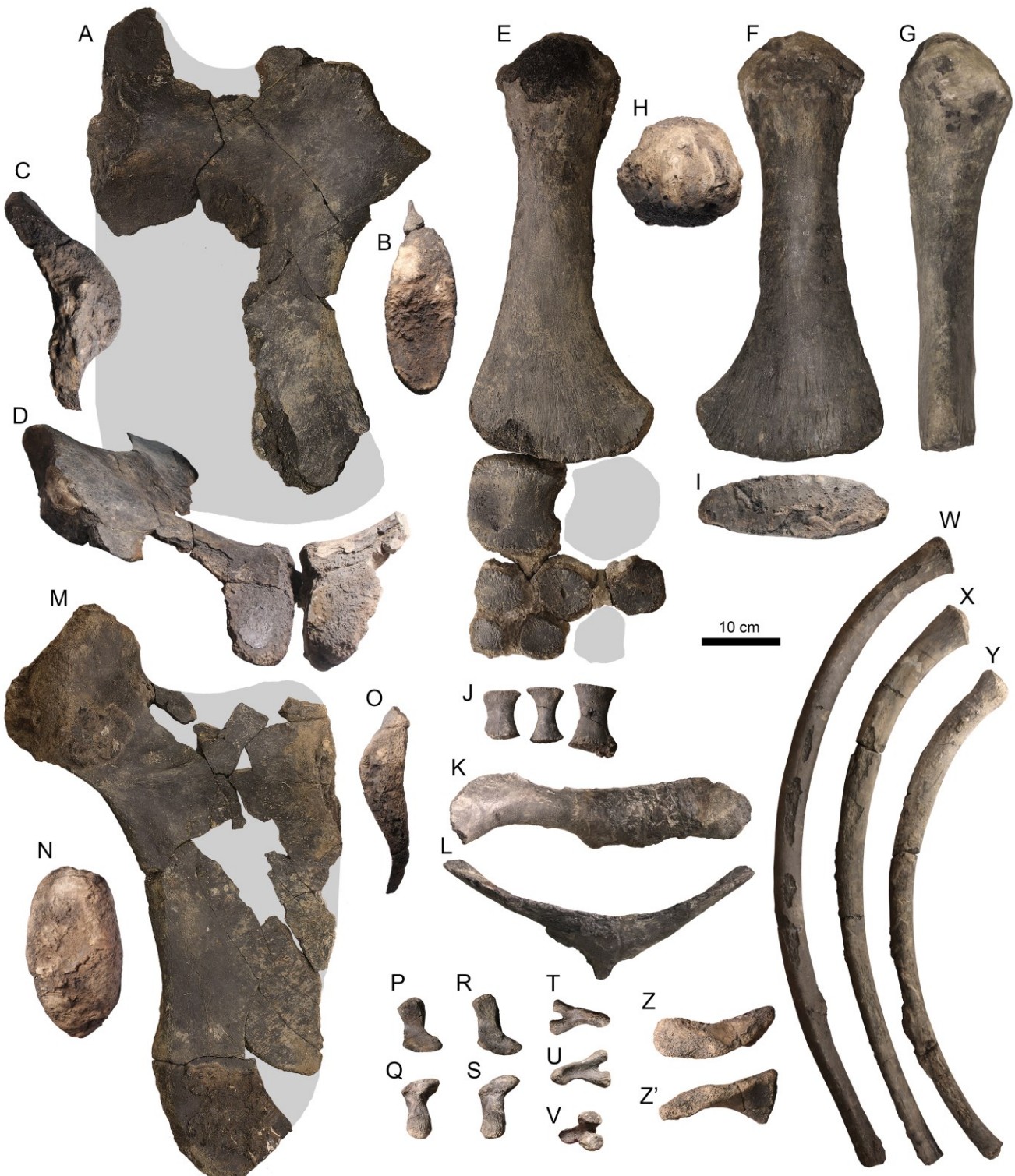

**Figure 6.** Postcranial skeleton of *Liopleurodon ferox*, SGM 1807. Right coracoid (**A–D**) in dorsal (**A**), glenoidal (**B**), mesial symphysial (**C**), and anterior (in articulation with the medial portion of left coracoid) (**D**) views. Partial right hindlimb in ventral view (**E**); right femur in dorsal (**F**), posterior (**G**), proximal (**H**), and distal (**I**) views. Phalanges (**J**). Interclavicle(?) in dorsal (**K**) and anterior (**L**) views. Left ischium in dorsal (**M**), acetabular (**N**), and mesial, symphysial (**O**) views. Cervical ribs (**P–V**) in dorsal (**P,R**), ventral (**Q,S**), anterior (**T**), posterior (**U**), and proximal (**V**) views. Dorsal ribs (**W–Y**) and sacral rib (**Z,Z′**). Reconstructed outlines of broken parts are shown in gray.

A partial right hindlimb is preserved, including the femur in articulation with the tibia and mesopodial elements, as well as several isolated phalanges (Figure 6E,J). The

femur is 535 mm long proximodistally and 238 mm wide anteroposteriorly at its distal end. The femur has a long shaft with almost parallel anterior and posterior sides for the essential part of its length (Figure 6E,F). The proximal portion is expanded to form a capitulum and dorsal trochanter (Figure 6H). The distal end bears two subequal facets for the tibia and fibula, making an obtuse angle to one another. The tibia is isometric, as long as it is wide (120 mm). It has a convex and tapered anterior edge and concave posterior edge that contribute to the extensive spatium interosseum between the tibia and fibula.

The distal expansion of the femur in SGM 1807 is less pronounced than that in any known Oxford Clay *Liopleurodon* specimen [5,117], which may suggest that SGM 1807 represents a distinct species of this genus.

### 5.1.2. *Simolestes* sp.

The left maxilla SGM 1574-1 is anteroposteriorly short and massive; it bears a small rounded external naris and its posterior edge is irregularly digitated for contacts with other cranial elements (i.e., frontal, prefrontal and lacrimal), which is typical for pliosaurids [118]. It lacks the orbital contribution known for rhomaleosaurids and plesiosauroids [119]. The posterior edge of the bone is incomplete ventrally. The preserved length of SGM 1574-1 is 260 mm, maximum dorsoventral height is 90 mm. There are nine alveoli on the preserved part, and several smaller alveoli could have been present on the broken posteroventral portion. The first two alveoli do not exceed 20 mm in diameter (Figure 7B), and the diameter of the subsequent alveoli increases abruptly, reaching 33 mm at the third alveolus. Starting from the sixth, the diameter of the alveoli decreases (Figure 7B). The body of the bone expands laterally in the region of the third to fifth alveoli, which had the largest caniniform teeth. The significant anteroposterior shortening, robustness, and circular outline of the external naris and alveolar pattern allow the referral of SGM 1574-1 to *Simolestes*, the only Callovian pliosaurid genus with such a short and robust snout [5,6,116].

The teeth of *Simolestes* are distinctive in having fine apicobasal enamel ridges, more numerous on the lingual surface of the crown [116]. The ridges on the labial surface always begin some distance from the base, which is a constant feature of the teeth of this taxon used in its recognition [5,6,116]. The largest teeth of *Simolestes* have no ornamentation on the labial surface of the crown; the individual enamel ridges are narrower, less raised from the surface of the enamel, and apparently do not coarsen upwards in comparison to *Liopleurodon*, the individual ridges can be irregularly roughened and wavy [116]. A tooth crown described by Bogolubov [33] from the middle Callovian of Gzhel greatly matches these characteristics, and we concur with Tarlo [6] in its referral to *Simolestes*. Two other teeth from the middle and upper Callovian of Moscow and Ryazan regions (PIN R-3589 from Gzhel and specimen in the private collection of V. Bakhtin from the Zmeinka Quarry, see [39]) match some, or all of, the above characteristics (Figure 8A–C); therefore, they are here referred to *Simolestes* sp.

Of interest is a large propodial, SGM 569-1 (Figure 7D–G), from the lower Callovian of the Unzha River. It is 58 cm long proximodistally, 31 cm wide anteroposteriorly at the distal end, and 11 cm thick distally. Its capitulum is largely broken off, so the complete length of the bone could have reached 60 cm. The great distal expansion of the propodial clearly marked off from the shaft allows the referral of SGM 569-1 to *Simolestes*, which shares such propodial configuration [5], differing it from other Middle and Late Jurassic pliosaurids [5,6]. If this referral is correct, SGM 569-1 is among the largest representatives of the genus, as it is 16% longer than the femur of the holotype (NHMUK PV R 3319).

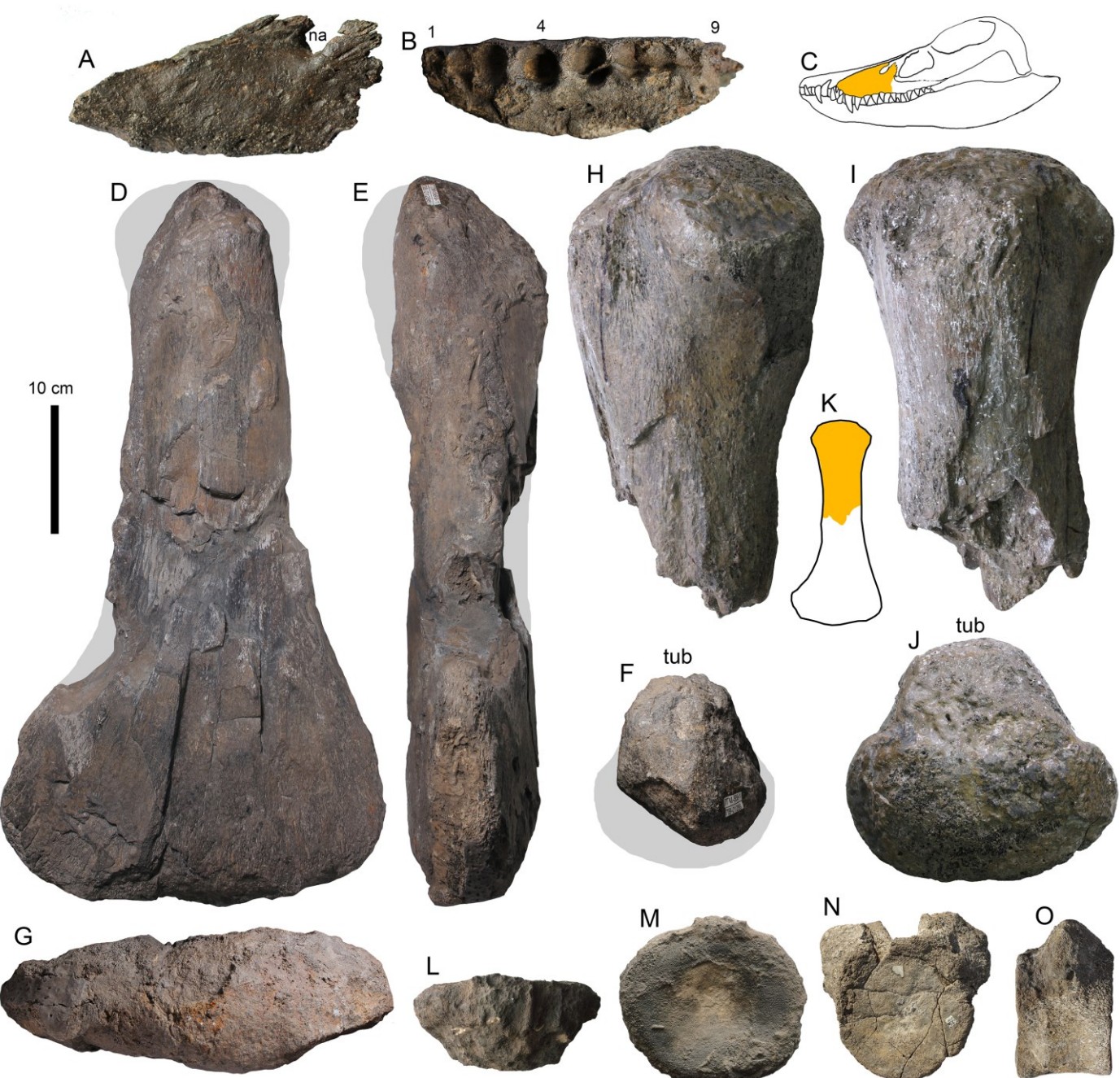

**Figure 7.** Pliosaurid remains. Left maxilla of *Simolestes* sp. SGM 574-01 (**A–C**), in lateral (**A**) and ventral (**B**) views. (**C**) Position of the element in the skull (in orange). Femur of *Simolestes* sp. SGM 569-1 (**D–G**), in dorsal (**D**), posterior (**E**), proximal (**F**), and distal (**G**) views. Proximal portion of a large propodial (likely femur), NNGASU 147/2080 (**H–K**), in anterior or posterior (**H**), dorsal (**I**), and proximal (**J**) views. (**K**) Position of the fragment (in orange). Articular portion of the vertebral centrum, YSPU M/F-45, in (?) dorsal (**L**) and articular (**M**) views. Pliosaurid sacral vertebra, TsNIGR 157a/649, in articular (**N**) and right? lateral (**O**) views. Abbreviations: 1–9, positions of maxillary alveoli; na, external nares; tub, dorsal trochanter/tuberosity.

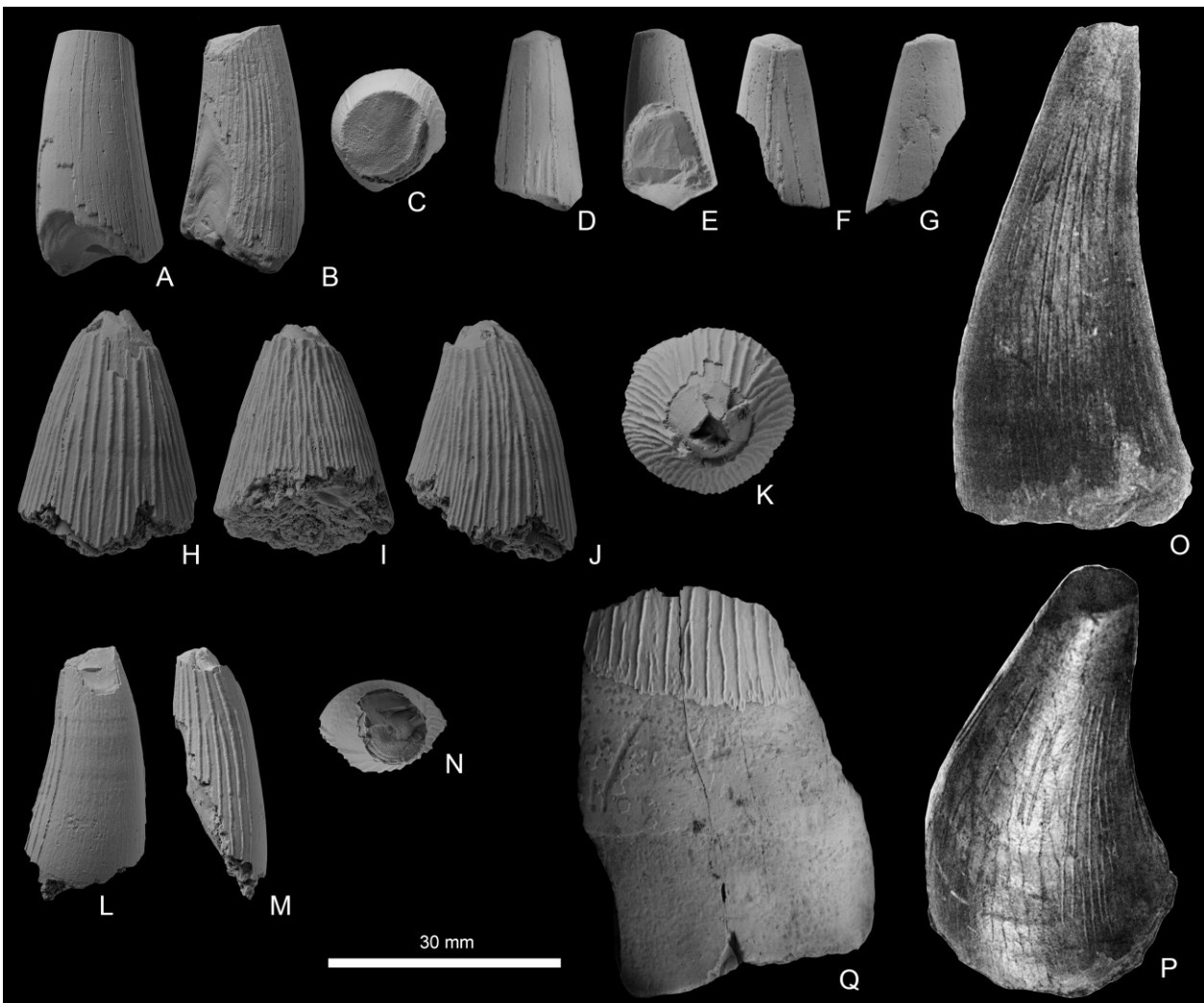

**Figure 8.** Pliosaurid teeth from the Callovian of European Russia. Pliosauridae cf. *Simolestes* PIN R-3589 (**A–C**) from the middle Callovian of Gzhel; Pliosauridae indet., PIN 5818/8 (**D–G**) from the middle Callovian of Rechitsy; Pliosauridae indet. PIN R-3574 (**H–K**) from the middle Callovian of Nikitino; Pliosauridae indet. PIN R-3575 (**L–N**) from the upper Callovian of Dubki; lost holotype of *"Thaumatosaurus calloviensis"* (**O,P**), referred to *Simolestes* sp. herein, from the middle Callovian of Rechitsy, reproduced from Bogolubov [33] (pl. II, figs. 1 and 6); Pliosauridae indet. SGM 2007-5 (Q) from the lower Callovian of Mikhalenino, Kostroma Region. Views: labial (**A,E,H,L**), lingual (**D,I**), mesial or distal (**B,F,G,J,M,P,Q**), apical (**C,K,N**), and oblique labial from the apex (**O**).

### 5.1.3. Pliosauridae gen. et sp. indet.

There are several specimens, represented by fragmentary tooth crowns and some postcranial remains attributable to pliosaurids, but not diagnostic at the genus level.

Tooth crown fragments from Dubki (PIN R-3575), Nikitino (PIN R-3574), Gzhel (PIN 5818/8), and Mikhalenino (SGM 2007-5) localities (Figure 8) do not show distinct features for referral to a particular pliosaurid genus, although the relatively large size of PIN R-3575 and PIN R-3574 and the semicircular cross-section of enamel ridges suggest the referral to *Simolestes*, rather than *Liopleurodon*.

NNGASU 147/2080 is the proximal portion of a very large propodial bone from the Kama River basin (Figure 7H–J). Its preserved length is 37 cm, and the diameter of the capitulum is 23 cm. The precise locality and age of this specimen is not recorded. Based on the museum label, it is Middle Jurassic in age, which implies the Callovian age, although the Bathonian age cannot be ruled out, considering the stratigraphy of the region, e.g., [71,114]. This specimen is equally large to the "Stewartby Pliosaur" NHMUK PV R

8322 from the *E. coronatum* Zone of the UK [120] and indicates the presence of large plio-saurids, around 8-9 m long, in the Middle Jurassic of European Russia.

A sacral vertebra TsNIGR 157a/649 (Figure 7N,O), from Yelatma, Ryazan Region, has moderate size with a 115 mm maximum diameter. A slightly larger (143 mm in maximum diameter) is the articular surface of a centrum, YSPU M/F-45, from the lower Callovian of the Makariev locality at the Unzha River (Figure 7L,M).

A partial postcranial skeleton MM 219 was described and figured by Riabinin [41], who somewhat conventionally referred it to *Peloneustes philarchus*. However, apart from the relatively small size for a pliosaurid, no distinct character supports this referral. There-fore, we follow the opinion of other researchers [43,44] and consider this specimen as Plio-sauridae indet.

*5.2. Plesiosauroids*

5.2.1. *Cryptoclidus eurymerus*

The specimen PIN R-3600 is a partial postcranial skeleton comprising several cervi-cal, pectoral, dorsal, and caudal vertebrae, rib fragments, some gastralia, fragments of gir-dle, and limb elements (Figures 9 and 10). This specimen originates from the middle Cal-lovian of Nikitino, Ryazan Region. It was excavated by the Club of Junior Paleontologists of the Paleontological Museum, Moscow, led by one of us (A.S.) in the summer of 2014 and 2015. This specimen can be positively referred to *Cryptoclidus eurymerus* based on an-terior expansion of the humeral distal portion bearing the radial facet and the respectively enlarged anterior expansion of the radius portion bearing the humeral facet, which is more than twice as long as the facet for the radiale (autapomorphies of *C. eurymerus* [7]). Neural arches fused with the centra and radii angular and tightly articulated with humeri indicate that the specimen is an osteologically mature individual [7]. This is also sup-ported by the size of preserved elements; the humerus is 34.5 cm long and the femur 33.5 cm long, which agrees with the dimensions of large *C. eurymerus* specimens reported by Andrews [4], and indicates the total length of PIN R-3600 being around 4.2 metres.

The preserved cervical and pectoral vertebrae have concave articular surfaces (com-monly referred to as amphicoelous type), oval in outline (Figure 10A,D, and G). The cer-vical and pectoral centra are mediolaterally wider than they are long anteroposteriorly and high dorsoventrally (Figure 10A–I). There are no ventral and lateral keels. The prezyg-apophyses are trough-like, with articular surfaces faced anteromedially, they are not fused medially and their combined width is narrower than the centrum width. The pre-served middle dorsal vertebra has circular in outline articular facets (Figure 10J), its trans-verse processes are raised high dorsally, and the long axis of the rib facet is nearly vertical (Figure 10K). When articulated with the rib, it results in a marked posterior inclination of the dorsal rib and a more dorsoventrally compressed body cross-section (Figure 9N) than that reconstructed by O'Keefe et al. [121] (fig. 7B). This indicates that *Cryptoclidus* was more similar to the Oxfordian *Tatenectes* in its low body shape than was previously thought [121].

From the forelimbs, the left humerus and both radii are preserved, although heavily weathered (Figure 9A–E). As noted above, the humeri and radii are diagnostic in *Crypto-clidus eurymerus* in forming a strong anterior protrusion of the leading edge of the fore-limb, which is well pronounced in PIN R-3600. In the left forelimb, the radius and hu-merus are so tightly articulated that they become partially fused anteriorly.

The left femur is nearly completely preserved, with its proximal portion only slightly weathered (Figure 9G,H), whereas the right femur is heavily weathered (Figure 9K). The femora are well consistent with the femora described and figured for *Cryptoclidus eu-rymerus* [4,7]. The femur has a slender and long shaft and is not as greatly expanded dis-tally as the humerus (Figure 9F–H). The distal end of the femur is dorsoventrally thicker than that of the humerus (45 vs. 37 mm).

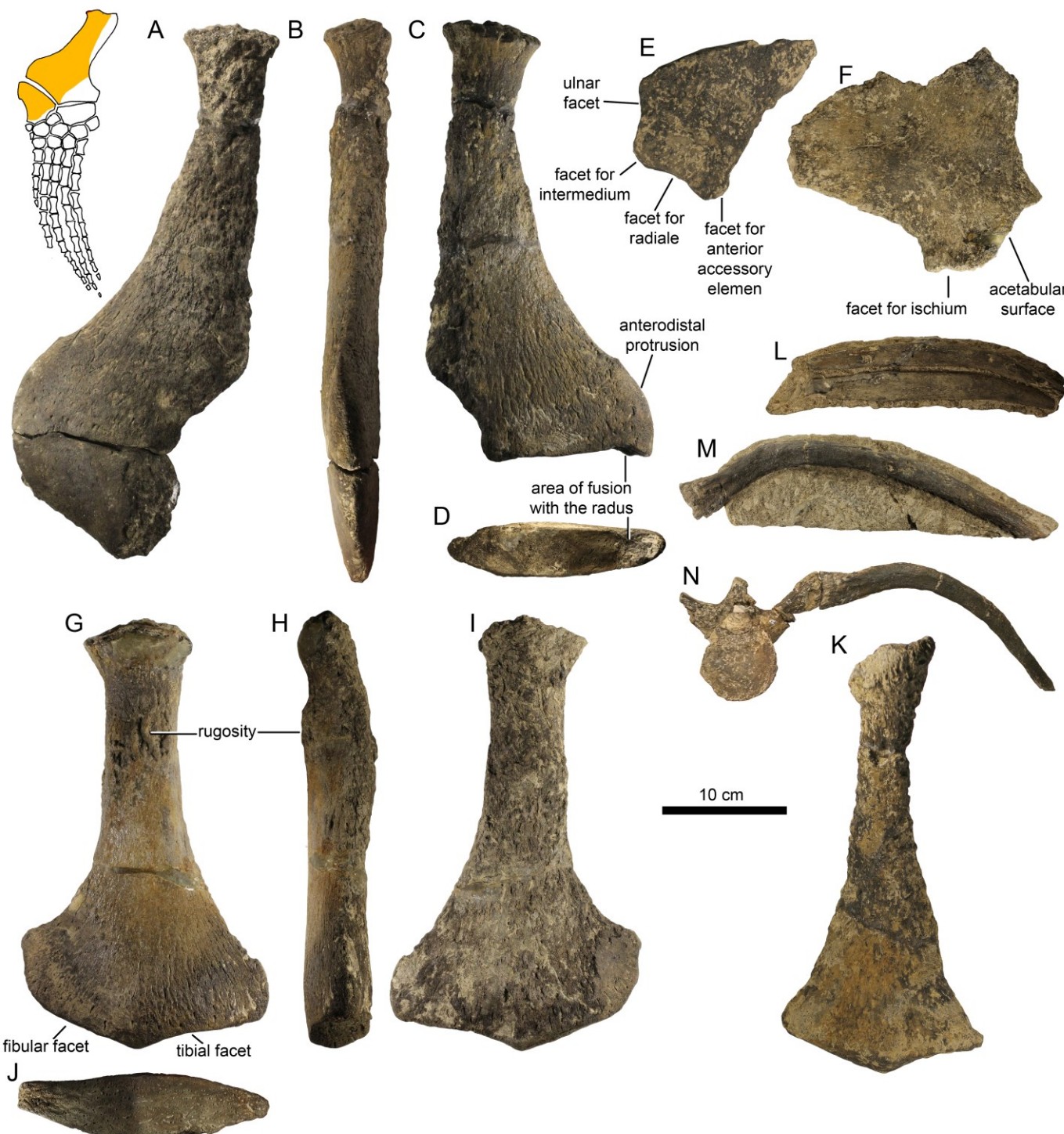

**Figure 9.** Postcranial elements of *Cryptoclidus eurymerus*, PIN R-3600, from the middle Callovian of Nikitino, Ryazan Region. Left humerus in articulation with the radius in dorsal (**A**) and anterior (**B**) views; reconstruction of their position in the limb (in orange), left humerus in ventral (**C**) and distal (**D**) views. Right radius in dorsal view (**E**). Partial pubis in dorsal view (**F**). Left femur in ventral (**G**), anterior (**H**), dorsal (**I**), and distal (**J**) views. Partial right femur in dorsal view (**K**). Two gastralia (**L**), dorsal rib in anterior view (**M**), and in articulation with dorsal vertebra (**N**).

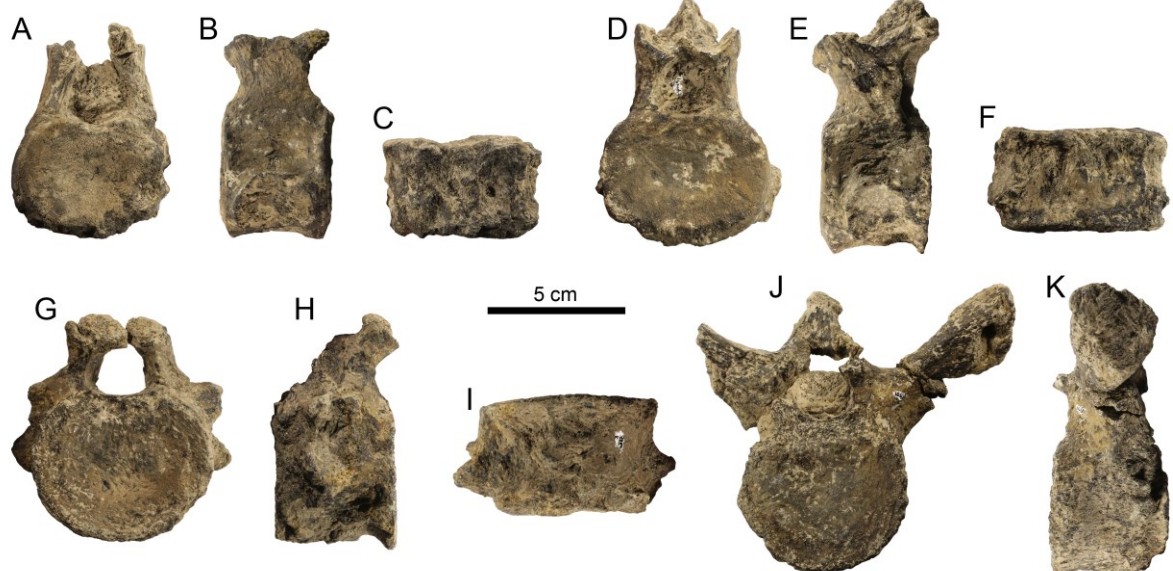

**Figure 10.** Vertebrae of *Cryptoclidus eurymerus* PIN R-3600, from the middle Callovian of Nikitino, Ryazan Region. Cervical vertebrae (**A–F**) in anterior (**A**,**D**), left lateral (**B**,**E**), and ventral (**C**,**F**) views. Pectoral vertebra (**G–I**) in anterior (**G**), right lateral (**H**), and ventral (**I**) views. Dorsal vertebra in anterior (**J**) and left lateral (**K**) views.

Other plesiosaurian specimens known from the Callovian of European Russia are represented by isolated bones (mostly vertebrae) and teeth. The teeth of the Oxford Clay cryptoclidids have been demonstrated to be diagnostic [7,122]. In the available plesiosaurian teeth from the Callovian of European Russia, three morphotypes occur, which are in agreement with the teeth of *Muraenosaurus*, *Cryptoclidus*, and *Tricleidus* ([7] and N.Z.'s pers. obs. on NHMUK specimens).

### 5.2.2. Isolated Plesiosauroid Teeth

Several teeth (SGM 2007-3/1–3/5; PIN 5819/3) are referable to *Muraenosaurus*. The largest of these (SGM 2007-3/1) has a tooth crown 38 mm high. The tooth crown is slender and gently curved, subcircular in cross-section, becoming more oval to the apex (Figure 11E,F,R). Its labial side is largely smooth, with a band of short and discontinuous ridges present at the base (Figure 11D,J,U). The lingual side is ornamented by numerous ridges, some of which extend almost to the apex, but none reach it (Figure 11B,H,O). The ridges are straight and sparse in the upper half of the crown; to the base, they became numerous and form a vermiculation pattern (Figure 11A,B,G,H,S,T). This morphology perfectly matches that described for *Muraenosaurus leedsii* by Brown [7] and among Callovian plesiosaurians found only in this species (*Cryptoclidus* has small teeth with rare ridges, *Tricleidus* has ridges on the labial surface [7]).

One small tooth crown, PIN R-3621, from the lower Callovian of Mikhaylovcement Quarry has a peculiar cross-section, slightly flattened on the labial side and undulating on the lingual side, due to the two apicobasal grooves separating the lingual surface onto three lobes. The ridges are rare; the two longest mesial and distal ridges separate the smooth labial surface from the ridged lingual side, and between them, there are ten fine ridges on the lingual side, none of which reach the apex (Figure 11M). This morphology perfectly matches that of *Cryptoclidus eurymerus*, described by Brown [7], and although Brown had not described the trilobate cross-section of the teeth in *C. eurymerus*, this is confirmed by our personal observations (N.Z.'s pers. obs, on NHMUK PV R3730, April 2019). Therefore, we refer this tooth crown to *Cryptoclidus* sp.

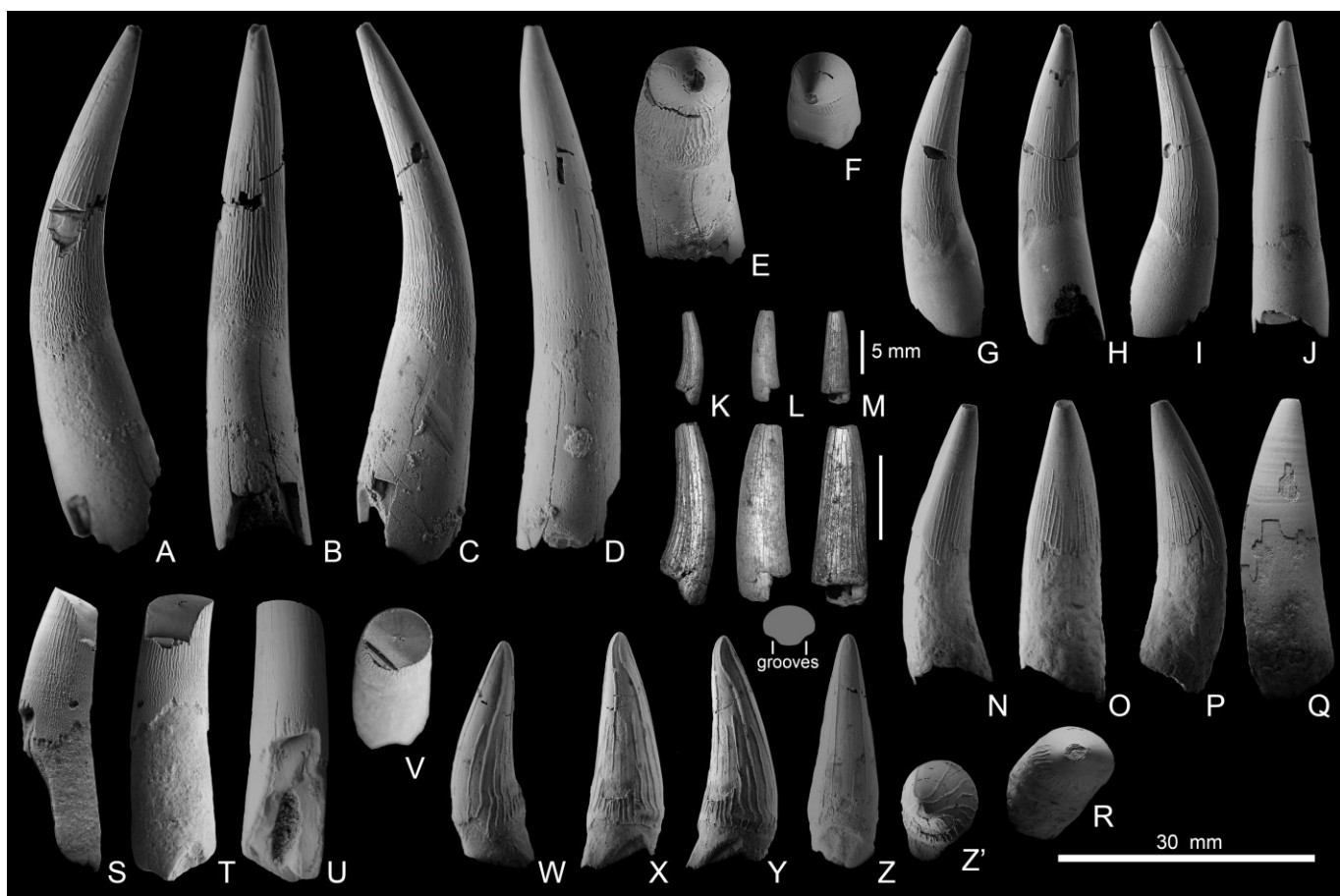

**Figure 11.** Isolated cryptoclidid teeth from the lower Callovian of Kostroma (**A–J**, **N–Z′**) and Ryazan (**K–M**) regions. *Muraenosaurus* sp. SGM 2007-3/1 (**A–E**), SGM 2007-3/2 (**F–J**), SGM 2007-3/3 (**N–Q**), PIN 5819/3 (**S–V**). *Cryptoclidus* sp. PIN R-3621 (**K–M**). cf. *Tricleidus* SGM 2007-4 (**W–Z′**). Views: mesial or distal (**A,C,G,I,K,N,P,S,W,Y**), labial (**D,J,L,Q,U,Z**), lingual (**B,H,M,O,X**), apical (**E,F,R,V,Z′**). Occlusal wear facets are visible at the tooth root on (**C**) and (**G,H**). K–M are SEM photographs with the bottom row magnified x2 relative to the top row and schematic cross-section of the crown with apicobasal lingual grooves shown below.

A tooth crown, SGM 2007-4, from the lower Callovian of the Unzha River differs from other teeth in its more robust ridges, not that sparse, as in *Cryptoclidus*, and not that frequent, as in *Muraenosaurus*. Six of the ridges reach the apex. The ridges on the labial surface are rarer and finer than those on the lingual side. Brown [7] provided restricted information on the teeth of *Tricleidus seeleyi*, and in his fig. 24 [7], the tooth of *T. seeleyi* appears superficially similar to that of *M. leedsii*. However, personal examination of the type specimen (N.Z.'s pers. obs, on NHMUK PV R3539, April 2019) confirms that the teeth of *T. seeleyi* differ from those of other Oxford Clay cryptoclidids by having more robust ridges, lack of dense vermiculation pattern at the base (characteristic of *M. leedsii*), the presence of sparse but prominent ridges on the labial side, and several (more than two) ridges reaching the apex (no ridges reach the apex in *M. leedsii* and only two reach the apex in *C. eurymerus*). All these features are present in SGM 2007-4, therefore, we refer it to as cf. *Tricleidus*. An interesting feature of SGM 2007-4 is that the ridges, reaching the apex, curve and form a spiral pattern (Figure 11X,Y,Z′). To our knowledge, such a condition has not been reported for plesiosauroids and may represent some abnormality of tooth development.

### 5.2.3. Isolated Plesiosauroid Vertebrae

We identify two morphotypes of plesiosauroid cervical vertebrae in the available material.

To Morphotype 1 (Figure 12) we refer vertebrae with flat or only slightly convex articular surfaces (platycoelous to acoelous type). Among Oxford Clay cryptoclidids, this condition is present in *Muraenosaurus* and *Picrocleidus* ([7] and N.Z.'s pers. obs.).

In *Muraenosaurus*, the articular surfaces are only slightly wider than they are high, and are semicircular in cross-section, whereas in *Picrocleidus*, the articular surfaces are markedly wider than they are high, oval in outline, and bear distinct depression under the neural canal floor [4]. Furthermore, *Picrocleidus* is a small taxon, with the longest cervical vertebral centra of the largest known specimen being 39 mm long [4], whereas most of the vertebral centra of Morphotype 1 are larger (44 mm and longer). Therefore, these vertebrae are referred to cf. *Muraenosaurus*. In the smaller anterior to middle cervical vertebrae, the anteroposterior length equals or slightly exceeds the mediolateral width (Figure 12B,E). In larger posterior cervical vertebrae, mediolateral width (W) exceeds the dorsoventral height (H) and anteroposterior length (L), so that $W > H > L$ (Figure 12G–T). There are no lateral and ventral keels. A slight depression on the dorsal side of the articular surface under the neural canal is present. The cervical ribs are dorsoventrally flattened at the base and fused with the centrum without visible sutures in all vertebrae of Morphotype 1 (Figure 12B,E,J,N,Q and T), indicating the maturity of the specimens (sensu Brown [7]). The neural arches are also fused to the centrum, however sutures are noticeable in some specimens (Figure 12B). At present, six vertebrae of Morphotype 1 are identified from the Callovian of European Russia, they all are depicted in Figure 12.

To Morphotype 2 (Figure 13A–H) we refer proportionally shortened ($W > H > L$) cervical vertebrae with markedly concave articular surfaces (amphicoelous type), which are known for Callovian cryptoclidid genera *Cryptoclidus* and *Tricleidus* [7], and for the Oxfordian cryptoclidid *Tatenectes* [123]; therefore, these vertebrae are referred to Cryptoclididae indet. There are only three isolated vertebrae of this morphotype. Two associated centra from the middle Callovian of Peski Quarry, PIN R-3591-3592, belong to a small, osteologically immature individual (Figure 13A–D), with their neural arches and ribs not fused to the centrum (criterion of immaturity, sensu Brown [7]). Specimen PIN R-3670, from Mikhaylovcement Quarry, belongs to a larger, osteologically mature individual with the neural arch and ribs fused to the centrum without any visible suture (Figure 13E–H). Furthermore, there are proportionally short pectoral and sacral vertebrae with concave articular facets from the Callovian of Mikhaylovcement Quarry (Figure 13U–B').

A small anterior dorsal vertebra (Figure 13I–K) was collected in association with the skeleton of *Liopleurodon* SGM 1807 and possibly represents its gastric contents. This vertebra has neural ach fused to the centrum, thus it belongs to an osteologically mature individual, although its small size (length 33 mm) indicates that it belonged to a small-bodied taxon, such as *Picrocleidus* or *Tricleidus* [4].

Another dorsal vertebra TsNIGR 144/1712 belongs to a large and osteologically mature plesiosaurian. Its articular surfaces are nearly flat, transverse processes are horizontal, and the long axis of the fib facet is nearly vertical (Figure 13M), all of which is characteristic of *Muraenosaurus* [4,121]. Therefore, we identify this specimen as cf. *Muraenosaurus*.

The dorsal vertebra YSPU M/F-44, in contrast, has concave articular surfaces and dorsally deflected transverse processes. These characters, together with the relatively large size allow the referral of YSPU M/F-44 to cf. *Cryptoclidus*.

Several isolated dorsal and caudal vertebral centra cannot be identified further than Plesiosauria indet. These are shown in Figure 13Q–T and C'–I'. The dorsal centrum, SGM 1891-05, belonging to a juvenile individual, is interesting in terms of its great anteroposterior shortness. A poorly preserved anterior caudal centrum, PIN R-3201, lacks chevron facets and has a rib facet confluent with the neural arch facet. A small posterior caudal centrum, PIN R-3596, has large chevron facets, and its rib and neural arch facets are also confluent.

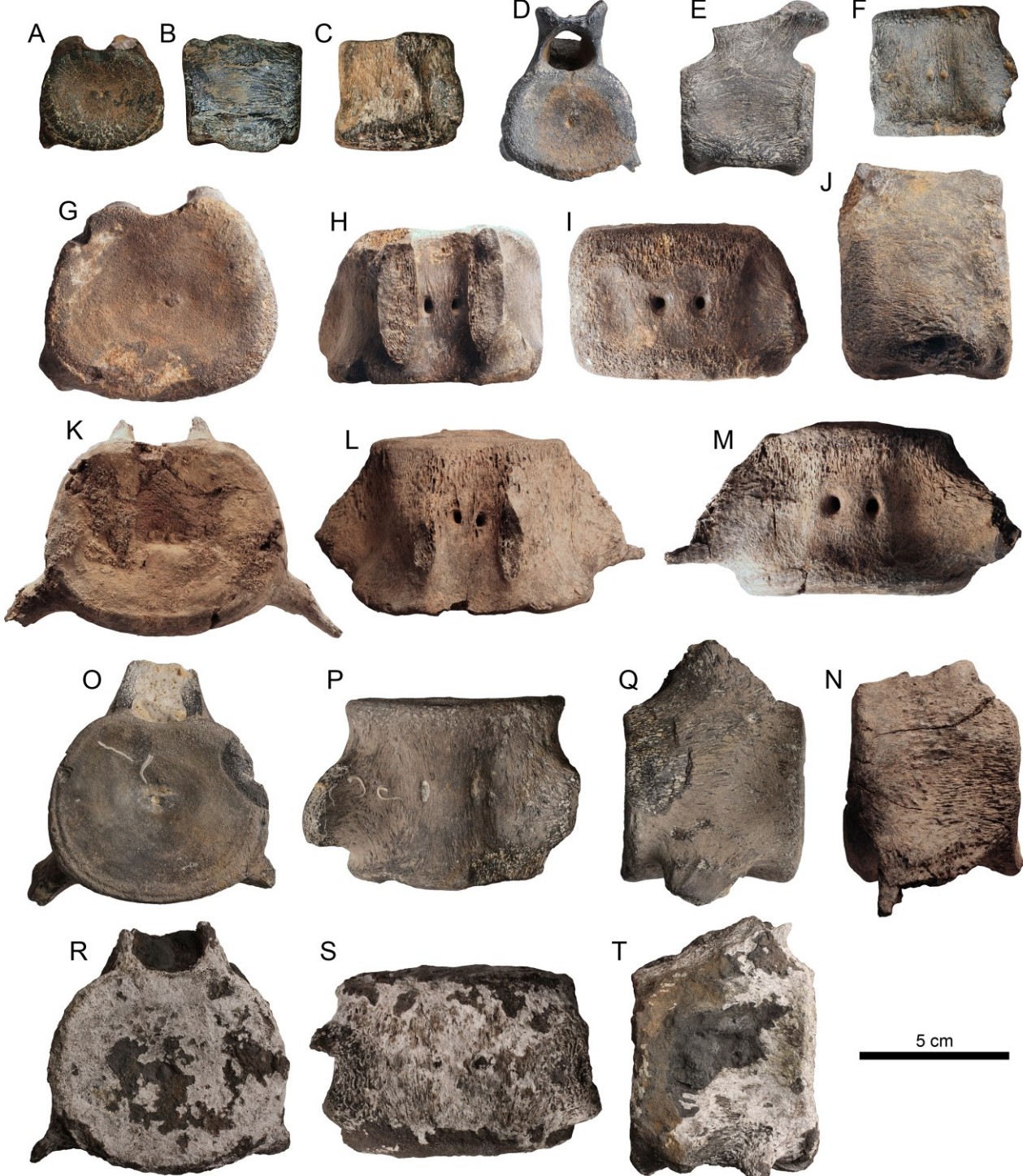

**Figure 12.** Cervical vertebrae of cf. *Muraenosaurus* (Morphotype 1). Anterior cervical vertebra SGM 1358-37 from the (?)lower Callovian of Alpatyevo (**A–C**), originally described by Bogolubov [33]. Anterior cervical vertebra SGM 1891-08 (**D–F**) from the lower Callovian of Mikhalenino. Posterior cervical vertebrae SGM 1891-10 (**G–J**) and SGM 1891-9 (**K–N**) from the lower Callovian of Mikhalenino. Middle to posterior cervical vertebra PIN R-3590 (**O–Q**) from the upper Callovian of Peski. Posterior cervical vertebra PIN R-3595 (**R–T**) from the middle Callovian of Mihailovcement. Views: anterior articular (**A,D,G,K,R**), lateral (**B,E,J,N,Q,T**), ventral (**C,F,I,M,P,S**), and dorsal (**H,L**).

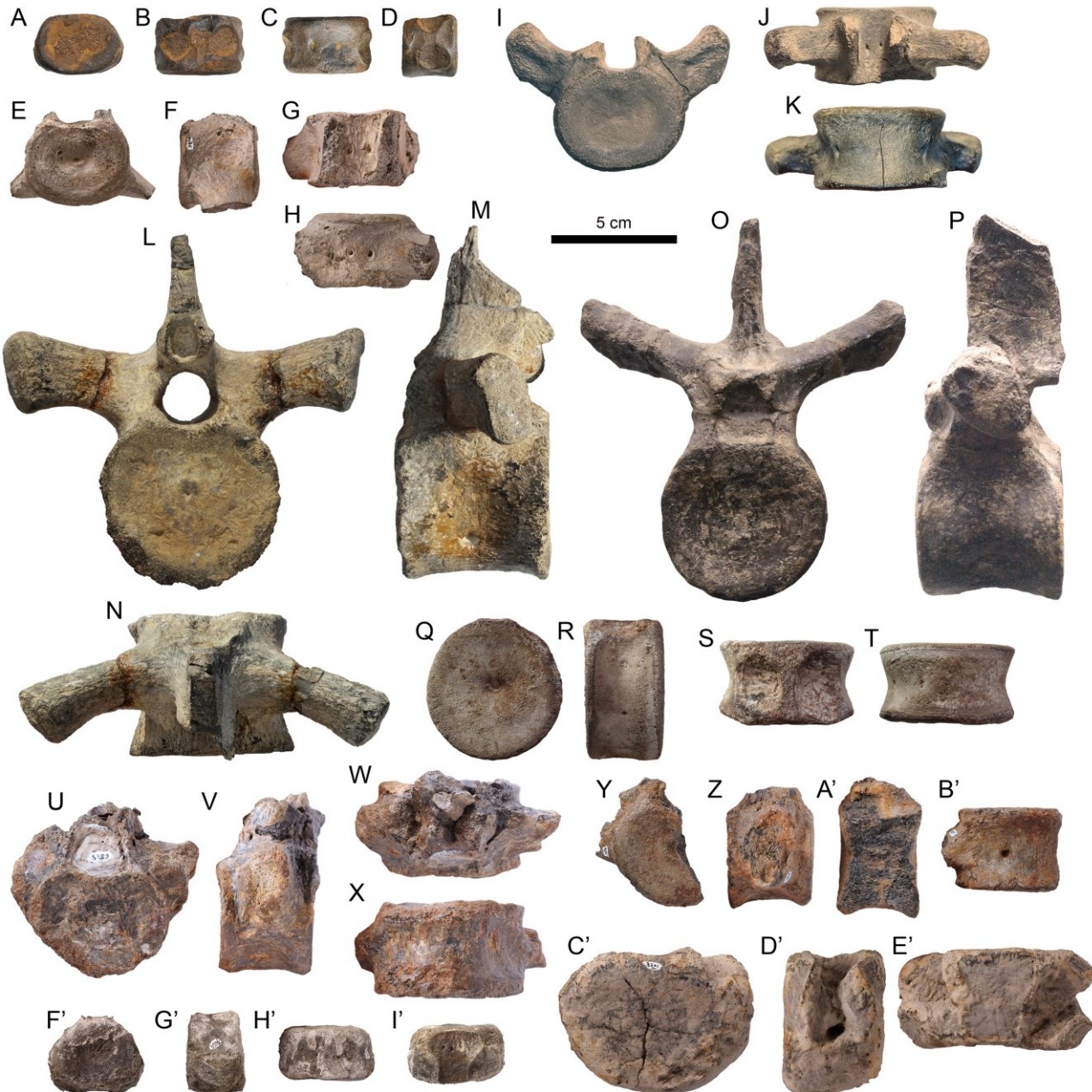

**Figure 13.** Plesiosauroid vertebrae from the Callovian of European Russia. Cervical vertebrae of short-necked cryptoclidids (Morphotype 2) (**A–H**); juvenile PIN R-3591 (**A–D**) and osteologically mature PIN R-3670 (**E–H**). Anterior dorsal vertebra of osteologically mature small cryptoclidid collected in association with *Liopleurodon* skeleton SGM 1807. Dorsal vertebra cf. *Muraenosaurus* TsNIGR 144/1712 (**L–N**). Dorsal vertebra cf. *Cryptoclidus* YSPU M/F-44 (**O,P**). Dorsal centrum of osteologically immature plesiosaurian SGM 1891-05 (**Q–T**). Pectoral PIN R-3709 (**U–X**), sacral PIN R-3606 (**Y–B′**), and caudal PIN R-3201(**C′–E′**) and PIN R-3596(**F′–I′**) vertebrae. Views: articular (**A,E,I,L,O,Q,U,Y,C′,F′**), dorsal (**B,G,J,S,W,E′,H′**), ventral (**C,H,K,T,X,B′,I**), lateral (**D,F,M,P,V,Z,D′,G′**), cross-section (**A′**).

*5.3. Ichthyosaurians*

5.3.1. Ophthalmosaurian Ichthyosaurs

The first documented finding of an ichthyosaur in the Callovian of Russia was made in 1925, near Rybkino Village (at present, the Republic of Mordovia), by a prominent Russian geologist and paleontologist, Alexander A. Stuckenberg (according to PSM archive records). The excavated specimen is represented by a nearly complete sclerotic ring, small

jaw fragments with associated teeth, several vertebral centra, fragmental neural arches, proximal portion of the humerus, and complete radius and ulna (Figure 14). However, this specimen, PSM 3999-4004, has not been described and figured until now. It is worth mentioning that in 2014, another skeleton referable to *Ophthalmosaurus* was excavated not far from this locality [38], in the lowermost Callovian strata, near Sinyakovo Village, Kras-noslobodsk District (Figure 14N). The preserved portions of the jaws of PSM 3999-4004 bear small teeth with roots circular in cross-section and slender crowns ornamented by rather sparse striations (Figure 14B–E). The crowns are 10–12 mm high and 5–7 mm at the basal diameter; they are very similar to those of *Ophthalmosaurus icenicus*, both in size and morphology [52]. The ulna and radius of PSM 3999-4004 are quite large (6 and 6.5 cm anteroposteriorly long, respectively), indicating a moderately large individual of approximately 3 m in length (estimated compared to known *O. icenicus* specimens; e.g., radius and ulna are each ~7–8 cm long in the holotype NHMUK PV R2133; N.Z.'s pers. obs.). The ulna of PSM 3999-4004 is proximally wider than the radius (Figure 14L). Its posterior edge is convex and demonstrates unfinished ossification, compared to the straight or concave and proximodistally long posterior edge of the ulna in *O. icenicus* from the Oxford Clay [52]. In this aspect, the ulna of PSM 3999-4004 is more similar to those of Late Jurassic *Arthropterygius* spp. [124], *Nannopterygius borealis* [125], and to the yet undescribed Bajocian ophthalmosaurian from Luxembourg [126]. The radius of PSM 3999-4004 has an oval outline with poorly demarcated facets, unlike a pentagonal outline in many ophthalmosaurids, including all the known specimens of *O. icenicus*, regardless of their ontogenetic state (N.Z.'s pers. obs. on numerous specimens in NHMUK and CAMSM collections, 2018–2019). This outline of the radius suggests the presence of two anterior surfaces, one for articulation with the preaxial accessory element and another free of contact with other elements. Among ophthalmosaurids, such a condition is present in *Ophthalmosaurus natans*, e.g., [127], some specimens of *Nannopterygius enthekiodon* [125], and *Ophthalmosaurus calloviensis* [35] (redescribed below). The sclerotic ring of PSM 3999-4004 (Figure 14A) is similar to that of many other ophthalmosaurids, with peripheral portions of the individual plates rather thin (Figure 14A). The preserved posterior caudal vertebrae are similar to those of *O. icenicus* [52], and the apical and fluke centra lack chevron facets (Figure 14H–K). In summary, despite the similarity of teeth to *O. icenicus*, the epipodial elements of PSM 3999-4004 are dissimilar to this species and resemble those of some other ophthalmosaurids, thus allowing its identification as Ophthalmosauridae indet. However, given its provenance and similarity to *O. calloviensis* from the Callovian of Saratov Region, PSM 3999-4004 may belong to this species, which is characterized below.

The first Callovian ichthyosaur that was formally described from Russia [35] is an incomplete forelimb, SSU104a/27, found in a trench near Dubki Village, Saratov Region. It was initially referred to a new genus and species, *Khudiakovia calloviensis*, by Arkhangelsky, 1999, and consequently synonymized with *Ophthalmosaurus icenicus* [50–52]. Arkhangelsky [53], however, retains this taxon as a valid species of *Ophthalmosaurus*. Our re-examination of this holotype specimen allows the definition of some difference to *O. icenicus* specimens from the Oxford Clay. With the humerus being 176 mm long, SSU104a/27 is a moderately large ophthalmosaurid within the size range of *Ophthalmosaurus* [4,52]. The humerus of SSU104a/27 is overall similar to that of *O. icenicus* [52] and lacks the constriction between the radial and ulnar facets known for *Arthropterygius* [124]. The presence of an accessory epipodial element, posterior to the ulna, suggested by Arkhangelsky [35], cannot be confirmed. However, the posterior edge of the ulna in SSU104a/27 is proximodistally short, convex, and has unfinished ossification; the radius of SSU104a/27 has an oval outline with poorly demarcated facets (Figure 15A,B), implying the presence of two anterior surfaces rather than one convex facet, as interpreted by Arkhangelsky [35]. Both the radius and ulna of SSU104a/27 are similar to those of the above-described PSM 3999-4004. The intermedium of SSU 104a/27 has two distal facets of subequal size for articulation with the fourth and third distal carpals, as in many ophthalmosaurids with "latipinnate" condition [52,124].

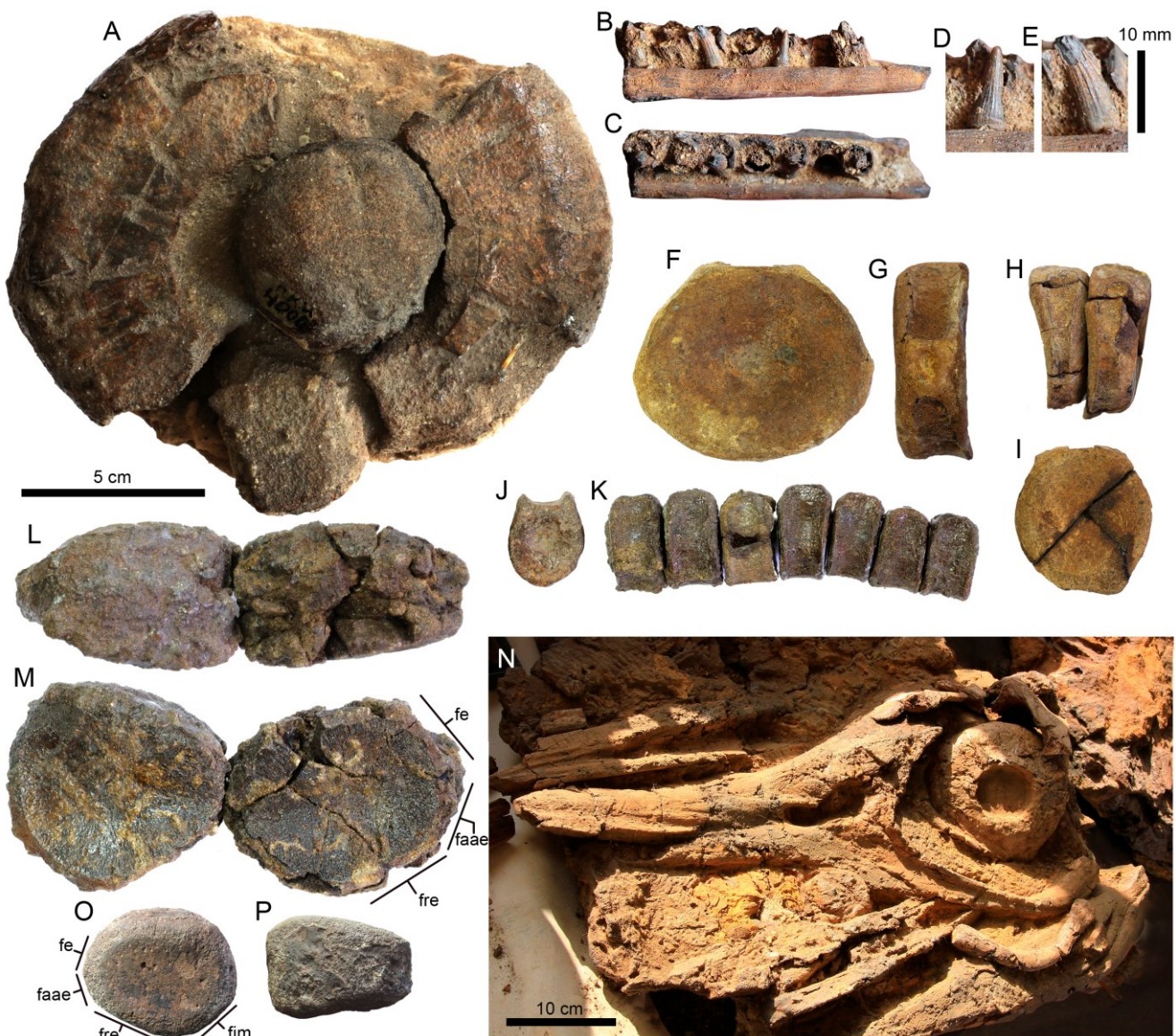

**Figure 14.** *Ophthalmosaurus* specimens from the lower Callovian of the Republic of Mordovia. Partial skeleton of *Ophthalmosaurus* cf. *calloviensis* PSM 3999-4004 (**A–M**) from Rybkino locality. Sclerotic ring (**A**). Dentigerous bone fragment (likely maxilla) in lateral (**B**) and ventral (**C**) views. Magnified teeth (**D,E**). Caudal preflexural (**F,G**), apical (**H,I**) and postflexiral, "fluke" (**J,K**), centra in articular (**F,I,J**) and lateral (**G,H,K**) views. Radius and ulna in proximal (**L**) and dorsal/ventral (**M**) views. Skull of a partial skeleton of *Ophthalmosaurus* sp. (**N**) from the lower Callovian of Sinyakovo locality reported by [38], in oblique anterolateral view. Isolated radius of *Ophthalmosaurus* cf. *calloviensis* (SGM 1891-06) from Trofimovshchina-2 locality in dorsal/ventral (**O**) and proximal (**P**) views. Abbreviations: faae, facet for anterior accessory element; fe, free surface; fim, facet for intermedium; fre, facet for radiale.

The morphology of the humerus in SSU104a/27 supports its referral to *Ophthalmosaurus*. However, the morphology of its epipodial elements questions the assignment to *O. icenicus* and demonstrates affinities to a wider range of ophthalmosaurids. The similarity of epipodial elements in SSU104a/27 and PSM 3999-4004 suggests that PSM 3999-4004 can represent the same taxon as SSU104a/27, for which the species name *O. calloviensis*, Arkhangelsky, 1999, is available. However, due to the fragmentary nature of both specimens, we consider *Ophthalmosaurus calloviensis* as species inquirenda, pending more complete materials.

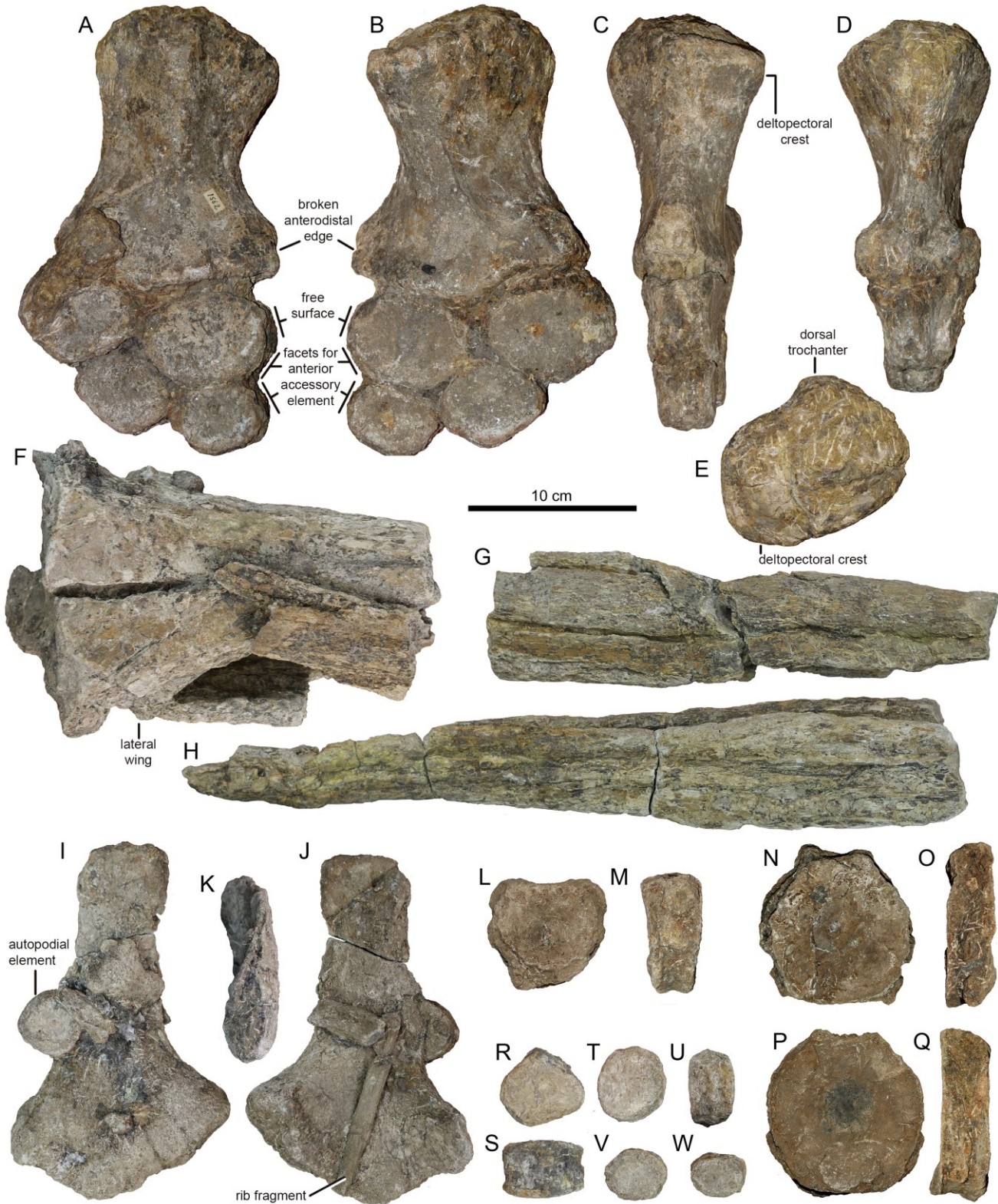

**Figure 15.** Ophthalmosaurids from the Callovian of Saratov Region. Holotype forelimb of *Ophthalmosaurus calloviensis* SSU 104a/27 (**A–E**) in dorsal (**A**), ventral (**B**), anterior (**C**), posterior (**D**), and proximal (**E**) views. Fragments of a partial skeleton SSTU MEZ 3/4 (**F–W**). Articulated nasals in dorsal view (**F**), partial premaxilla (**G**) and dentary (**H**) in lateral views. Right scapula (with associated rib fragments and distal limb elements) in dorsal (**I**), ventral (**J**), and proximal (**K**) views. Anterior dorsal (**L,M**), and posterior dorsal to anterior caudal (**N–Q**) centra in articular (**L,N,P**) and lateral (**M,O,Q**) views. Intermedium in dorsal/ventral (**R**) and proximal (**S**) views. Anterior accessory element in dorsal/ventral (**T**) and posterior (**U**) views. Phalanges in dorsal/ventral view (**V,W**).

An isolated radius, SGM 1891-06 (Figure 14O,P), from the lower Callovian (*P. elatmae* Zone) of the Trofimovshchina-2 locality, Republic of Mordovia, is also similar to the above-described specimens in being oval in dorsal view and having two anterior surfaces, further supporting this feature a as characteristic of early Callovian ophthalmosaurians of the Middle Russian Sea.

Another specimen of interest is an incomplete and poorly preserved skeleton, SSTU MEZ 3/4, found from the lower Callovian near Gvardeyskoe Village, Saratov Region (Figure 15F–W). The specimen has "lateral wings" over the external nares (Figure 15F), tapered anterior extremity of the dentary (Figure 15H), a scapula with a well-developed acromial process and short, mediolaterally compressed shaft (Figure 15I,J), intermedium rhomboid in dorsal view bearing two distal facets subequal in size (Figure 15R), and dorsoventrally thickened and rounded phalanges (Figure 15S–W). All this makes it possible to attribute the specimen to cf. *Ophthalmosaurus*.

In May 2022, a partial skeleton, SGM 1961, referable to *Ophthalmosaurus icenicus*, was discovered from the middle Callovian of the Perebory locality, Yaroslavl Region. The skull is poorly preserved and fragmented. Among the available parts, the most informative is the condylar portion of the left quadrate (Figure 16A–C), which has a shape and size typical for the Oxford Clay *O. icenicus* specimens. Vertebral centra from all regions of the spinal column are preserved, including the atlas–axis complex (Figure 17A–E). They also agree with the vertebral morphologies described for *O. icenicus* [52]. Among the interesting traits is that the third centrum has diapophysis and parapophysis fused into the dorsoventrally elongated synapophysis (Figure 17C,F), which is probably a malformation. Of interest is that the surface of the floor of the neural canal is roughened, bearing undulating longitudinal striations (Figure 17I,M,P and T). This condition is present in all Oxford Clay *O. icenicus* specimens, but is not seen in any other ophthalmosaurian (N.Z.'s pers. obs.), thus it likely represents an autapomorphy of *O. icenicus*. In every aspect, the humerus is similar to that of the *O. icenicus* paratype, NHMUK PV R2134, although the anterodistal edge in SGM 1961 is broken. The proximodistal length of the humerus is 14 cm, which corresponds to the size of young adult *O. icenicus* specimens [4]. Epipodial and autopodial elements are all typical for *O. icenicus*; the ulna is proximodistally elongated with a straight and dorsoventrally compressed posterior edge, the radius has a pentagonal outline dorsally, the intermedium bears two distal facets of subequal size, and the distal autopodial elements are rounded and dorsoventrally thick.

Another partial forelimb, PIN R-4956 (Figure 18I–N), from the upper Callovian of Mikhaylovcement Quarry, with the humerus 16 cm in proximodistal length, belongs to a larger individual. The humerus is nearly completely preserved and is indistinguishable in its shape from the *O. icenicus* NHMUK PV R2134 paratype humerus. Most noticeable is the anteriorly acute, triangular in outline facet for the preaxial accessory epipodial element (Figure 18N). This condition occurs only in *O. icenicus* and allows for the robust referral of the specimen to this species [128]. Associated with the humerus, the radius, intermedium, and a carpal element are also typical of *O. icenicus* in their shape and proportions.

The isolated premaxilla, SGM 2000-1 (Figure 19P,Q and R), is 50 cm long and 6 cm in maximum height. It has a characteristically anteriorly tapered and acute tip, which diverges from the sagittal plane anteriorly (Figure 19P). This anterior divergence of the premaxillae was reported to be unique to *O. icenicus* among Jurassic ichthyosaurians [52].

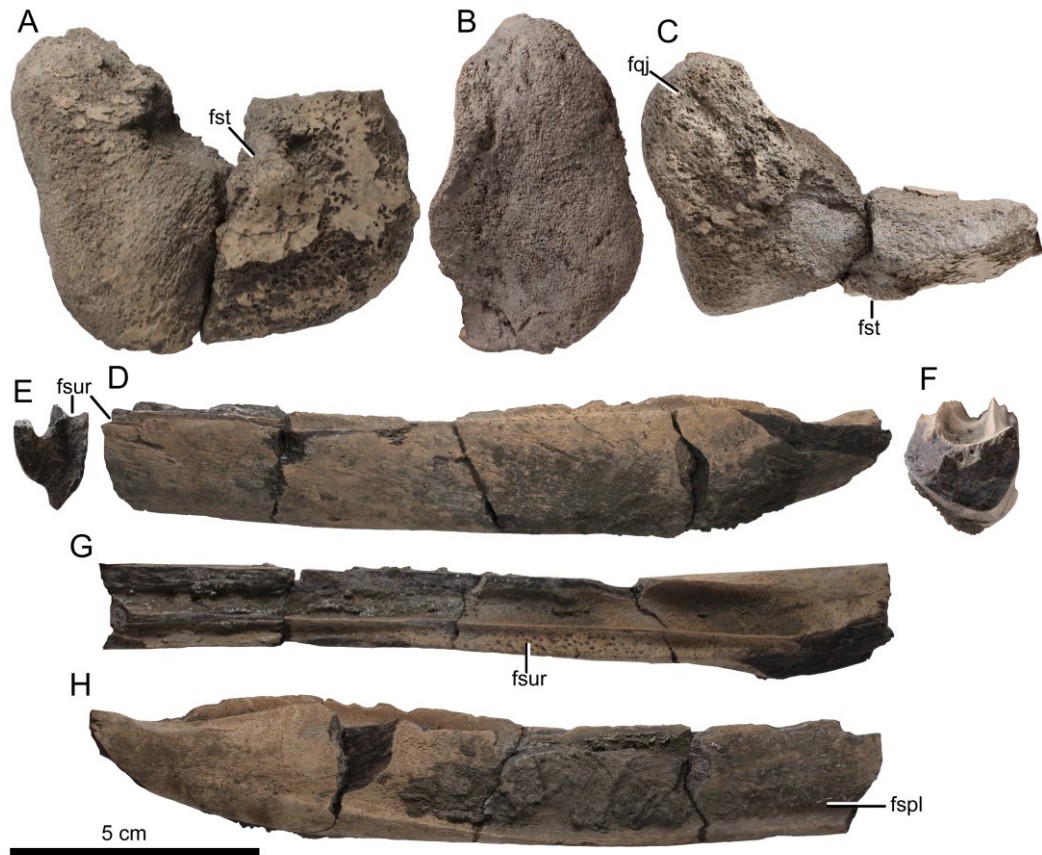

**Figure 16.** Cranial elements of *Ophthalmosaurus icenicus*, SGM 1961, from the middle Callovian of Perebory, Yaroslavl Region. Ventral portion of the left quadrate in posteromedial (**A**), condylar (**B**) and dorsal (**C**) views. Partial left angular in anterior cross-sectional (**E**), lateral (**D**), posterior (**F**), dorsal (**G**), and medial (**H**) views. Abbreviations: fqj, facet for quadratojugal; fspl, facet for splenial; fst, facet for stapes; fsur, facet for surangular.

Some other less complete specimens are also attributable to ophthalmosaurids and likely belonged to *Ophthalmosaurus*. Among these are several specimens from the lower Callovian of the Unzha River, including a partial mandible with teeth (SGM 1891-20; Figure 19A–D), and collected in association with it, an anterior dorsal centrum (SGM 1891-02; Figure 20A–C). The teeth are small (crowns 14 mm in maximum height and 7 mm at basal diameter) with slender curved crowns (Figure 19B–D), and weak roots of semicircular cross-section (Figure 19A–D), similar to *Ophthalmosaurus* [52] and *Arthropterygius* [124]. The anterior dorsal centrum has undulating longitudinal striations on the neural canal floor, which implies its affinity to *O. icenicus* (see above). The jugal SGM 1891-22 (Figure 19N,O) is a slender J-shaped element with anteroposteriorly narrow dorsal portion, very similar to that of *O. icenicus* [52]. The isolated left quadrate (SSU uncatalogued; Figure 19K–M) is also most similar in its morphology to *O. icenicus*, see [4,52]. An isolated tooth (SGM 2007-8; Figure 19E–H) and tooth crown (PIN 5819/4; Figure 19I,J) are small and slender, similar to the above-described teeth of SGM 1891-20 and PSM 3999-4004.

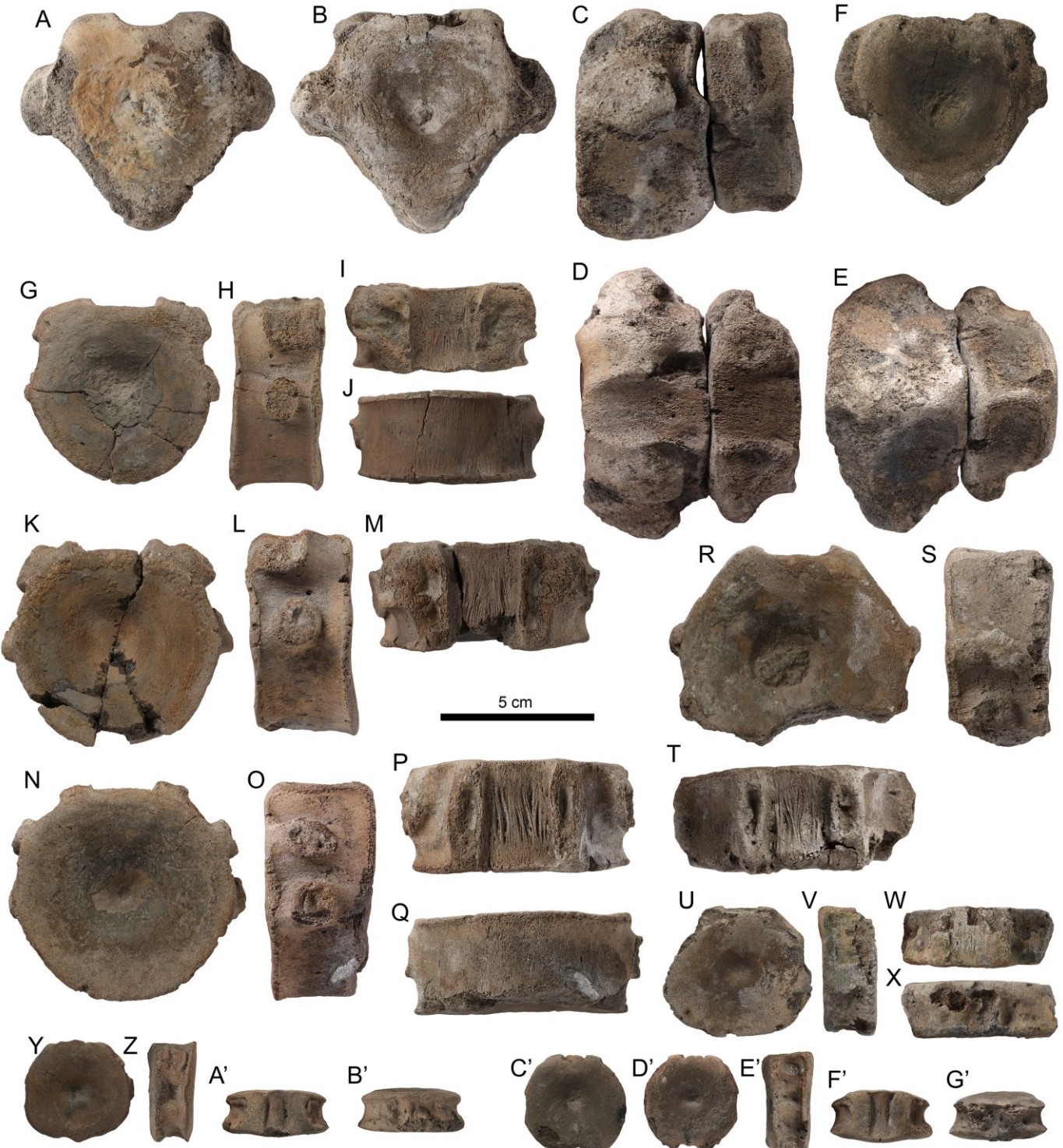

**Figure 17.** Vertebral centra of *Ophthalmosaurus icenicus*, SGM 1961, from the middle Callovian of Perebory, Yaroslavl Region. Atlas–axis complex in anterior (**A**) and posterior (**B**) views; articulated with the third centrum in left lateral (**C**), dorsal (**D**), and ventral (**E**) views. Third centrum in anterior articular view (**F**). Anterior dorsal (**G–M**), posterior dorsal (**N–T**), and caudal (**U–G′**) centra, in articular (**G,K,N,R,U,Y,C′,D′**), lateral (**H,L,O,S,V,Z,E′**), dorsal (**I,M,P,T,W,A′,F′**), and ventral (**J,Q,X,B′,G′**) views.

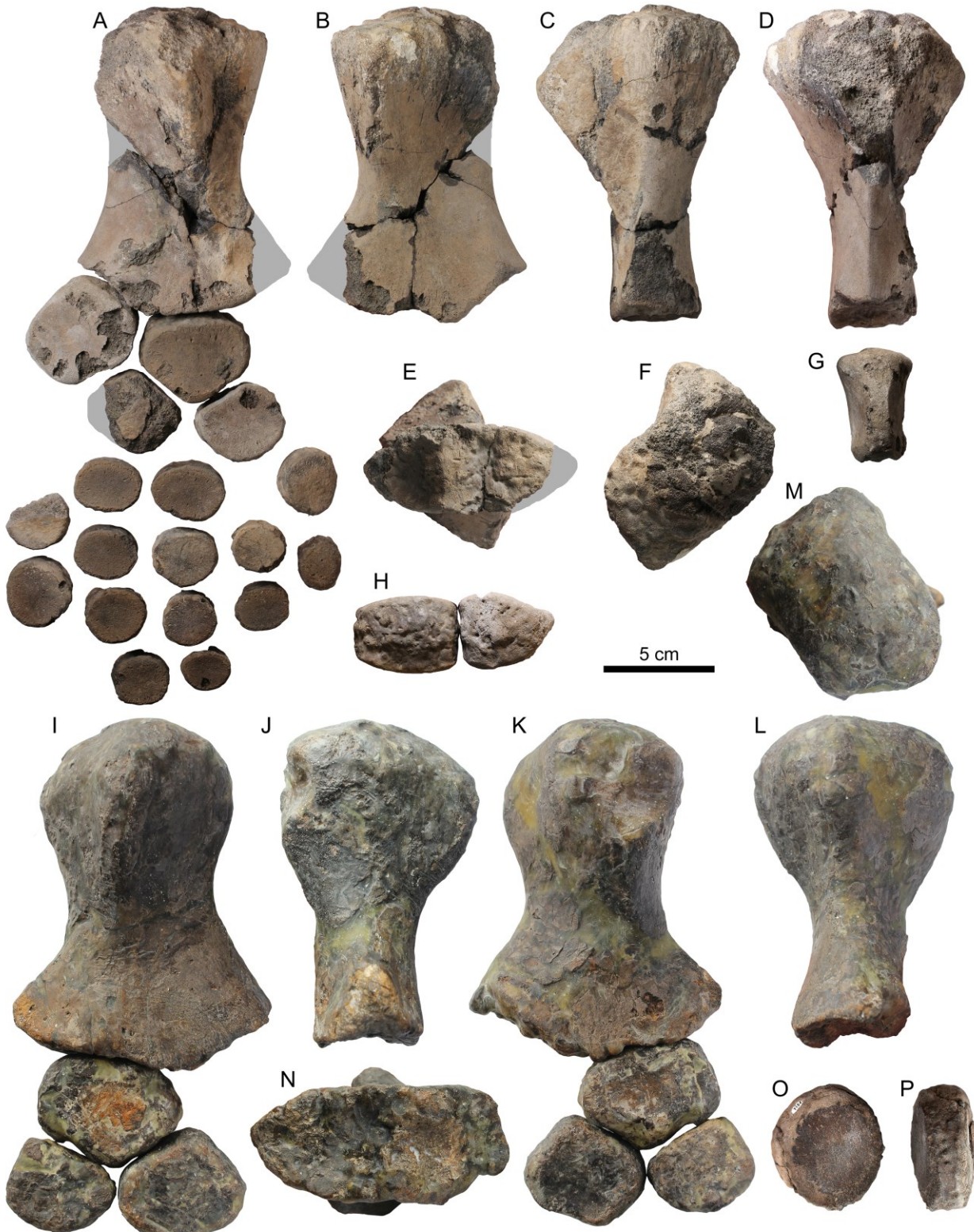

**Figure 18.** Forelimb remains of *Ophthalmosaurus*. Partial right forelimb of *Ophthalmosaurus icenicus*, SGM 1961, in dorsal view (**A**). Humerus of SGM 1961 in ventral (**B**), anterior (**C**), posterior (**D**), distal (**E**), and proximal (**F**) views. Ulna of SGM 1961 in posterior view (**G**), and proximal surfaces of the ulna and radius (**H**). Partial left forelimb of *Ophthalmosaurus icenicus* PIN R-4956 (**I–N**) from the upper Callovian of Zmeinka Quarry, Ryazan Region, in dorsal (**I**) and ventral (**K**) views. Humerus of PIN R-4956 in anterior (**J**), posterior (**L**), proximal (**M**), and distal (**N**) views. Anterior accessory epipodial element (?) of cf. *Ophthalmosaurus* PIN R-2516 (**O,P**), from the upper Callovian of Mikhaylovcement Quarry, Ryazan Region in dorsal/ventral (**O**) and anterior (**P**) views.

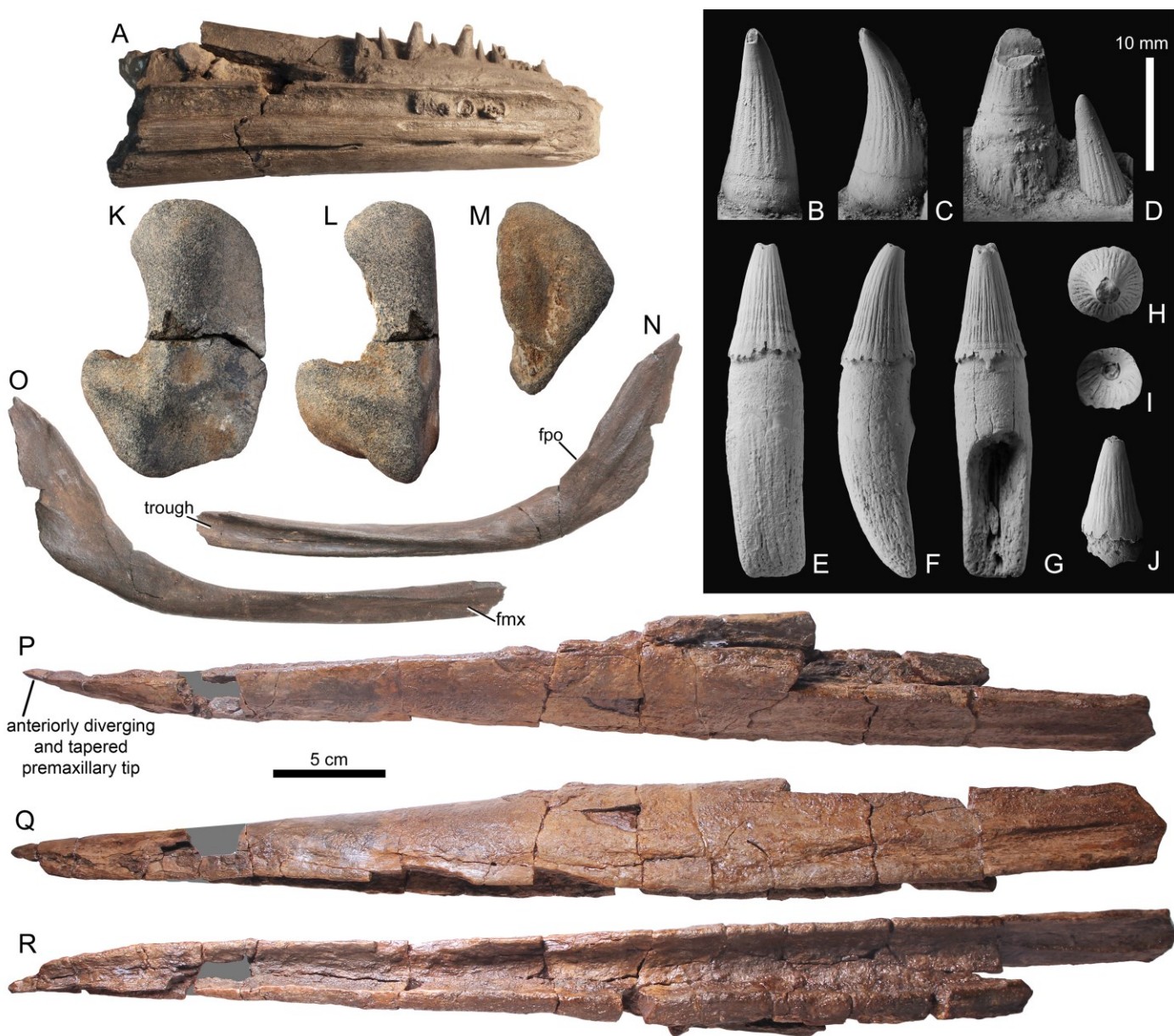

**Figure 19.** Ophthalmosaurid cranial remains from the Callovian of the Unzha River basin, Kostroma Region. Partial mandible with teeth SGM 1891-20 (**A–D**), on (**B–D**) magnified teeth. Isolated tooth SGM 2007-8 (**E–H**) and tooth crown PIN 5819/4 (**I,J**) in labial (**E,J**), anterior or posterior (**F,C**), lingual (**B,G,D**), and apical (**H,I**) views. Left quadrate, SSU uncatalogued, (**K–M**) in posteromedial (K), posterior (**L**), and condylar (**M**) views. Left jugal SGM 1891-22 in lateral (**N**) and medial (**O**) views. Left premaxilla SGM 2000-1 (**P–R**), in dorsal (**P**), lateral (**Q**), and ventral (**R**) views.

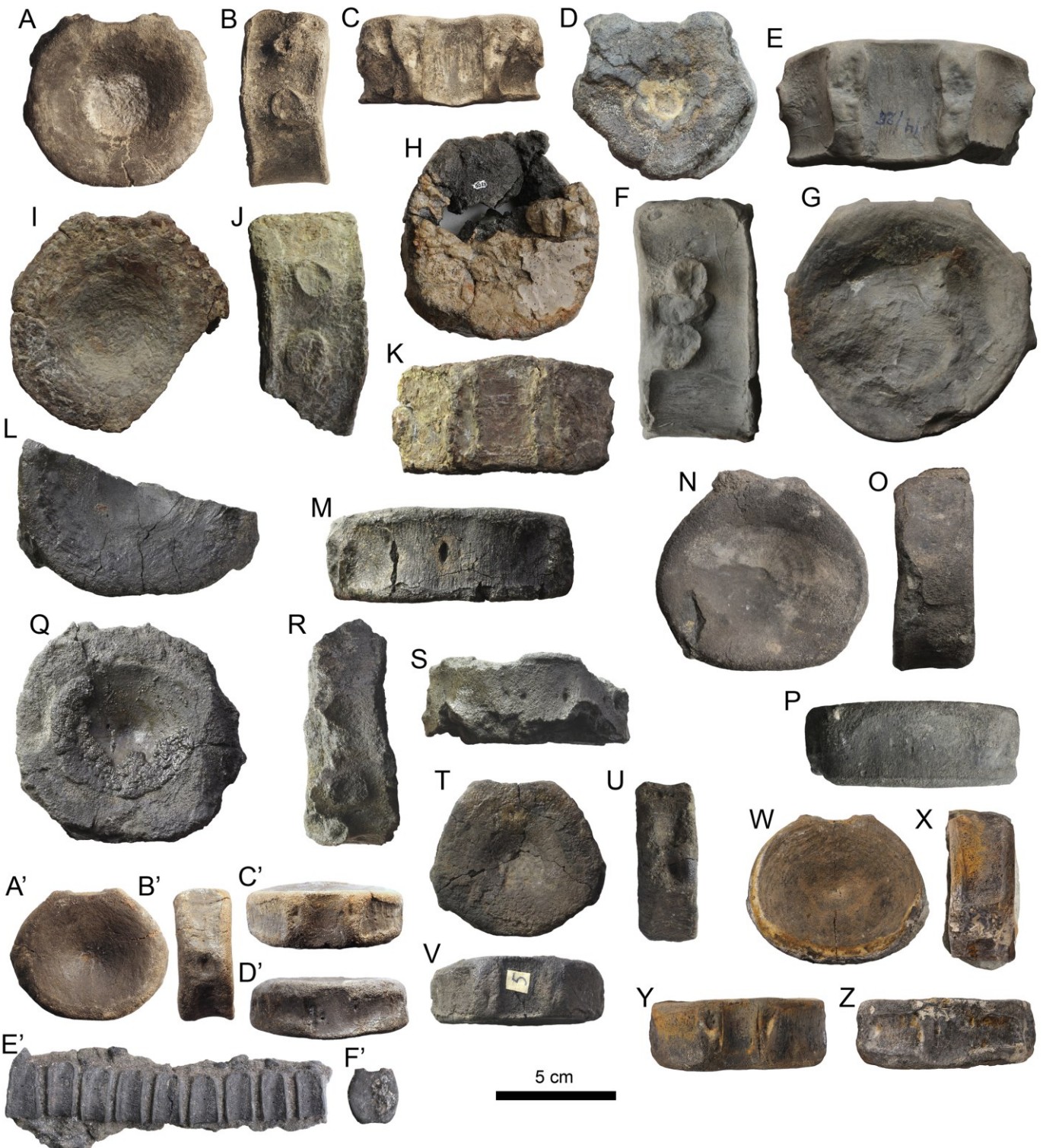

**Figure 20.** Ichthyosaurian veretebral centra from the Callovian of European Russia. Anterior dorsal centra SGM 1891-02 (**A–C**) and IG 93/13 (**D**). Middle dorsal centra ZIN PH 1/215 (**E–G**) and SGM 1891-04 (**I–K**). Posterior dorsal centra PIN R-3200 (**H**), SGM 1891-25 (**N–P**), SGM 1891-15 (**Q–S**). Anterior caudal centrum SGM 1891-19 (**L,M**). Posterior preflexural caudal centra SGM 1891-14 (**T–V**), PIN R-3593 (**W–Z**) and SGM 1891-24 (**A′–D′**). Series of postflexural (fluke) vertebrae SGM 1891-21 (**E′,F′**). Views: articular (**A,D,G,H,I,L,N,Q,T,W,A′,F′**), lateral (**B,F,J,O,R,U,X,B′,E′**), dorsal (**C,E,K,V,Y,C′**), ventral (**M,P,S,Z,D′**).

An isolated limb element, PIN R-2516 (Figure 18O,P), from the upper Callovian of Mikhaylovcement Quarry is rounded in outline and compressed at one of the edges. It likely represents an anterior accessory element, or one of the elements from the anterior or posterior digits of the limb. In its rounded outline and large size (6 cm in diameter), it is similar to the respective elements of *Ophthalmosaurus* [52] and *Arthropterygus* [124].

5.3.2. Indeterminate Ichthyosaurians

Numerous isolated ichthyosaurian vertebrae are known from the Callovian strata of European Russia. Some of these are the only finds from certain localities, including the northernmost occurrence in European Russia, Adzvavom [110] (see Table 1 for details). Some of these specimens are depicted in Figure 20. We omit the description of these specimens here, as in the present state of knowledge, it is not possible to identify them more precisely than Ichthyosauria indet., although they most likely belong to ophthalmosaurians. Therefore, these specimens only demonstrate the ubiquitous presence of ichthyosaurians in the Middle Russian Sea during the Callovian.

*5.4. Thalattosuchian Crocodylomorphs*

Recently Young et al. [40] described isolated thalattosuchian teeth from the Callovian of European Russia, therefore, we refer the reader to that contribution for details. Young et al. [40] identified three teeth from the lower Callovian of Unzha Village as Metriorhynchidae indet., and one more crown, from nearby Mikhalenino Village, as Geosaurini indet. (NB localities "Unzha" and "Mikhalenino" may represent different records of one natural outcrop near the villages Unzha, Mikalenino, and Popovo). A tooth crown from the lower Callovian of CHP-5 locality in Saratov, Saratov Region, originally described and figured by Arkhangelsky [35], was referred to Geosaurini indet., although it represents a taxon different to the Unzha geosaurin. One tooth crown from the lower Callovian of Gumny village, Republic of Mordovia, and another from the middle Callovian of the Mikhaylovcement Quarry, Ryazan Region, were referred to cf. *Thalattosuchus*. Two crowns from the middle Callovian of Gzhel and Mikhaylovcement were referred to *Tyrannoneustes* sp. [40].

In autumn 2023, a cervical vertebra (SGM 2007-02) from the Callovian of Mikhaylovcement Quarry was donated to SGM by Konstantin Volkov. The vertebral centrum (Figure 21A–F) has approximately equal height and width (50 and 49 mm, respectively), and its length (53 mm) only slightly exceeds the width. The anterior articular surface is flat to slightly convex, and the posterior articular surface is concave. The ventral hypapophyseal keel is well-developed and somewhat convex ventrally (Figure 21D). It is narrowest in the central part and expands into triangular surfaces anteriorly and posteriorly. These surfaces have a conspicuously rugose, mammilated surface (Figure 21E). The parapophyses are protruding and oval in outline, located closer to the anterior surface. The anterior surfaces of parapophyses are rugose (Figure 21D). The subequal length and width of the centrum suggest its referral to a geosaurine metriorhynchid, as metriorhynchine metriorhynchids have posterior cervicals that are slightly shorter than wide ([78]; M.T. Young pers. comm. 2023).

Furthermore, in SGM collection, there is a long bone from the lower Callovian of the Unzha River basin (Figure 21G–K), which likely represents a thalattosuchian metacarpal or metatarsal. It is a slender, 65 mm long element with a triangular to teardrop cross-section.

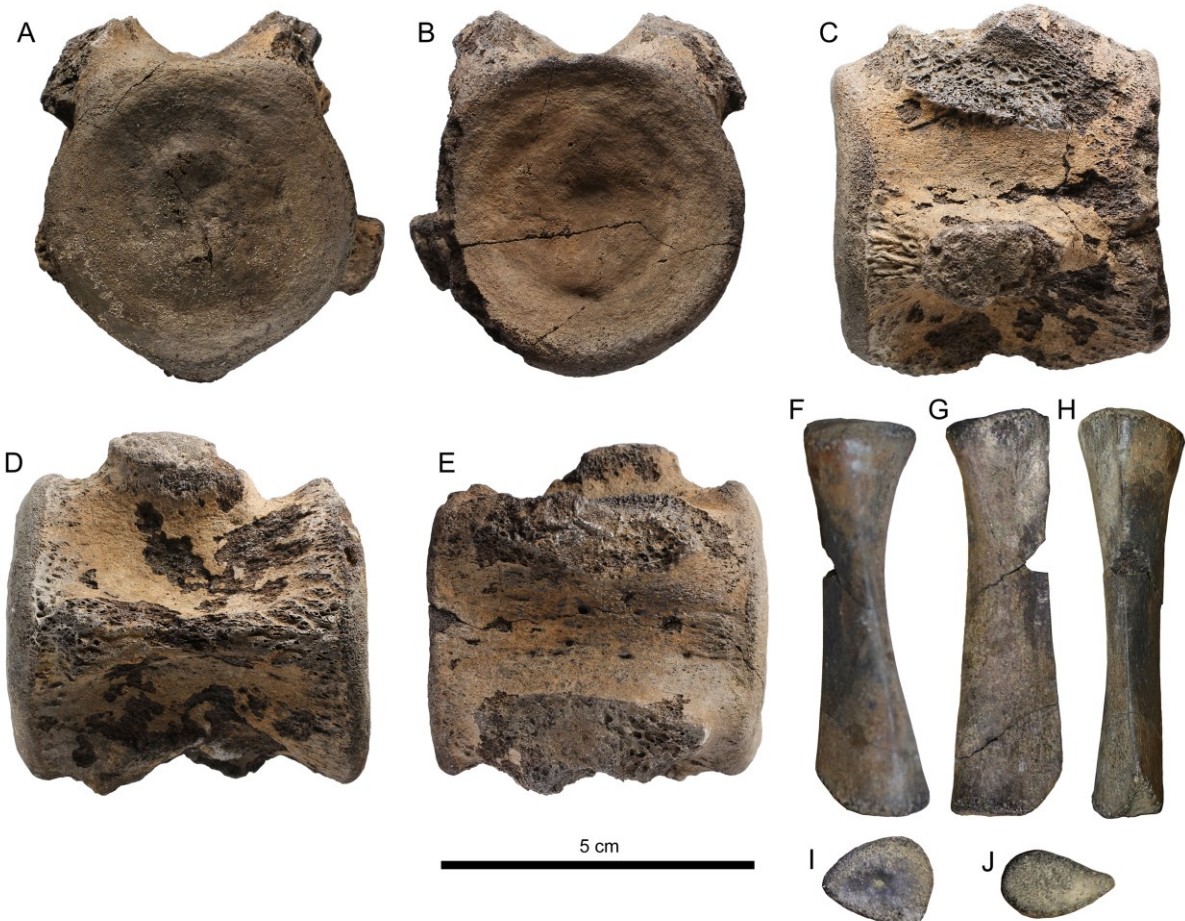

**Figure 21.** Metriorhynchid cervical vertebra SGM 2007-02 from the Callovian of Mikhaylovcement Quarry in anterior (**A**), posterior (**B**), left lateral (**C**), ventral (**D**), and dorsal (**E**) views. Thalattosuchian metacarpal or metatarsal SGM 1891-01 (**F–J**) from the lower Callovian of Mikhalenino.

## 6. Discussion

The fossil record of marine reptiles since the middle Toarcian and to the middle Callovian is extremely poor globally [1,2,126,129–136], which makes it hardly possible to assess any turnovers during this time interval, or trace patterns of marine reptile distribution [126]. Furthermore, our knowledge of marine reptile fauna of the only well-characterized middle Jurassic stage, the Callovian, is largely based on the Oxford Clay Formation of England, with less sound records from coeval strata of other countries of Western Europe [12–26], and extremely limited data on Callovian marine reptiles outside of Europe [27–32]. Moreover, for many Callovian marine reptile specimens from the Oxford Clay, precise information on their stratigraphic position was not recorded [4,5,137–139]. According to the existing records and later collections, the majority of marine reptile remains from the Oxford Clay Formation, particularly articulated skeletons and bone associations, appear to come from the basal beds of the Peterborough Member (Middle Callovian, lower part of the *Kosmoceras jason* Subzone of *K. jason* Zone; Figure 22) [8,139–141]. This is partially due to the peculiarities of clay extraction, both manual and mechanical, such that *Gryphaea* beds and concretionary nodules, especially abundant in Bed 10 [141], as well as the poorer quality of these basal beds due to frequent shells, resulted them becoming the predominant area for fossil collecting [139], even though a few finds from the below (uppermost lower Callovian *Catasigaloceras enodatum* Subzone of *S. calloviense* Zone [9]) and above (middle to upper Callovian *Erymnoceras coronatum*, *Peltoceras athleta* and *Lamberticeras lamberti* zones [8,83,139]) intervals of the Oxford Clay Formation are also known. Therefore, the taxonomically rich Callovian herpetofauna of England largely characterizes a short

episode in the Middle Callovian. Brown and Keen [122] also described the marine reptile fauna from the uppermost lower Callovian (*C. enodatum* Subzone of *S. calloviense* Zone) sands of the Kellaways Formation of the UK. They identified *Liopleurodon*, *Cryptoclidus*, *Muraenosaurus*, as well as teleosaurid and metriorhynchid teeth. Thus, they demonstrated that at least some reptile genera of the Oxford Clay were present already in the lower Callovian Kellaways Sands. Early Callovian marine reptile fauna of European Russia provides additional data for this poorly characterized time interval and for older intervals within the Callovian. Similarly to Brown and Keen [122], we identified *Liopleurodon*, *Muraenosaurus*, and *Cryptoclidus* in the lower Callovian of Russia, but also documented *Simolestes* and cf. *Tricleidus* herein. Thus, the lower Callovian plesiosaurians of European Russia are similar to those from the Kellaways and Oxford Clay formations of the UK, at least at the genus level. The fragmentary nature of the available lower Callovian specimens in both the UK and European Russia complicates further comparisons of these faunas with each other and with younger Oxford Clay fauna. However, it is possible that new findings will reveal some anatomic differences to characterize the temporal and/or geographic separation of the early Callovian marine reptile fauna of the Middle Russian Sea.

Of interest is the presence of a relict rhomaleosaurid in the lower Callovian of Russia, which is in line with the finds from the lower Callovian of Arctic Canada, as well as Callovian records from Argentina and the UK [36]. This probably shows that early Callovian marine reptile faunas particularly retained stratigraphically older taxa from poorly characterized faunas of the Aalenian–Bathonian [126].

Among metriorhynchids, Young et al. [40] reported several indeterminate metryorhynchid teeth, cf. *Thalattosuchus*, and two morphotypes of Geosaurini indet. from the lowermost Callovian of Russia. The tooth crowns referred to Geosaurini represent two distinct morphotypes, different from known European taxa, and imply that thalattosuchian fauna in the earliest Callovian of European Russia was partially different from that of the middle Callovian of Western Europe.

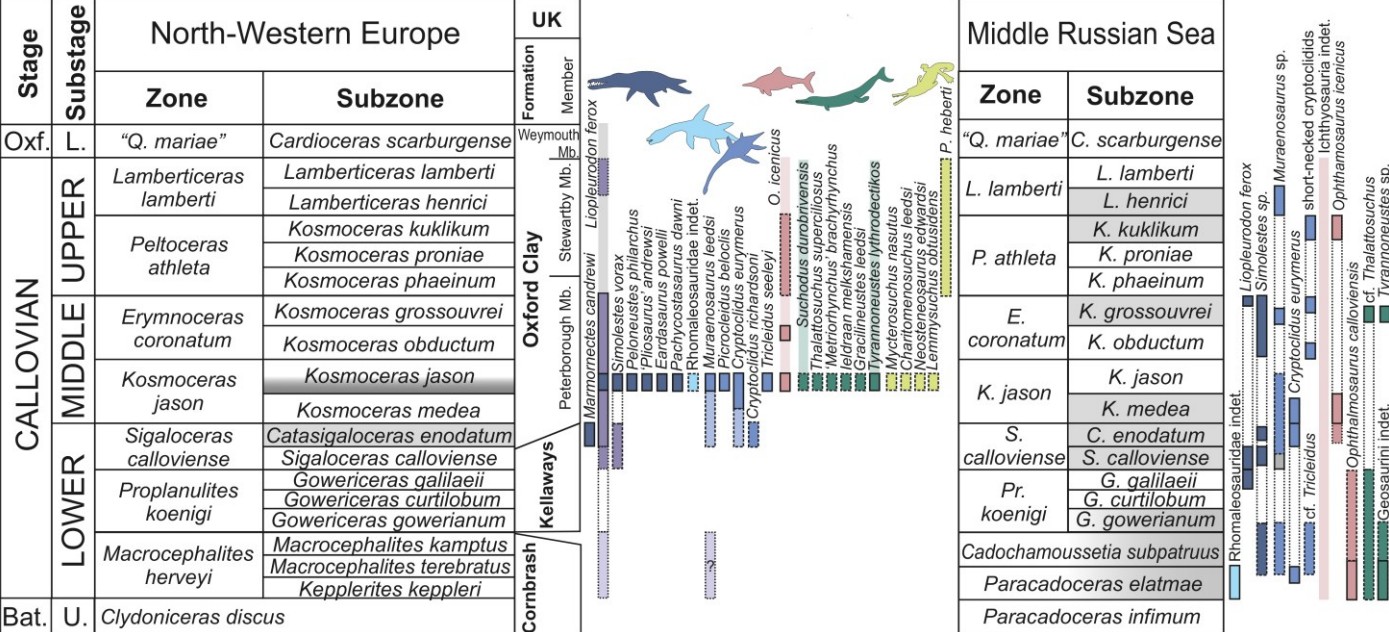

**Figure 22.** Stratigraphic distribution of marine reptiles in Western Europe (**left**) and European Russia (**right**). Ammonite zones/subzones highlighted in gray are the main levels with marine reptile fossils. For Western Europe, data on stratigraphic distribution of reptiles follow [8,9,73,122,139,142], and partially follow [78] and [143] for thalattosuchians. Occurrences with some uncertainties are shown in boxes with dashed margins; light shades indicate potential ranges based on ambiguous evidence or uncertainties in the age of specimens. Some ranges for early and late Callovian plesiosaurians in Western Europe are expanded according to Bardet [19,20] and Sachs and Nyhuis [25].

Similarly, early Callovian ophthalmosaurians of European Russia demonstrate some differences from the only known Callovian ophthalmosaurian taxon, *Ophthalmosaurus icenicus*, from the middle Callovian of Western Europe; whereas diagnostic specimens of middle to late Callovian ichthyosaurians in European Russia are referable to *O. icenicus*, supporting the connection of herpetofaunas of the Middle Russian Sea and seas of Western Europe. For this time interval, we also observe other typical representatives of the Oxford Clay herpetofauna in the Middle Russian Sea; plesiosaurians *Cryptoclidus eurymerus*, *Liopleurodon ferox*, *Simolestes* sp., and metriorhynchids cf. *Thalattosuchus* and *Tyrannoneustes* sp. [40], all of which have no marked differences from their contemporary congeners of Western Europe. This is not surprising, as the invertebrate faunas of the middle and late Callovian are also highly similar in these basins [91,144]. However, the studied area is still quite close to Western Europe on a global scale, and further data on Callovian marine reptiles from other regions of the world, especially those showing significant differences in invertebrate fauna, are required.

Some differences between the early Callovian ichthyosaurian and thalattosuchian taxa from European Russia and the known Callovian taxa of Western Europe could be explained by the older age of the Russian fauna, rather than its geographic position. However, the presence of several common genera implies that younger middle–late Callovian herpetofaunas inherited most of it. Thus, the earliest Callovian marine reptiles of European Russia expand our knowledge of the global diversity of marine reptiles of this age and are likely to represent the fauna ancestral to that of the middle Callovian age. Post-Callovian marine reptile faunas of Europe are less well known, as reptile remains are very rare and fragmentary in the Oxfordian of European Russia and Western Europe [145]. In this respect, it is still difficult to trace the patterns of marine reptile fauna evolution and geographic distribution during the Middle and early-Late Jurassic epochs in Europe and globally, and the Callovian remains a narrow, 4-million-year-long "window" into the Middle Jurassic marine reptile world, surrounded by the Aalenian–Bathonian and Oxfordian intervals of blurred marine reptile fossil records.

**Author Contributions:** Conceptualization, N.Z., M.A., and A.S.; Methodology, investigation, resources, data curation, all authors; writing—original draft preparation, N.Z. (most text), D.G. and A.I. (sections on paleobiogeography and stratigraphy); writing—review and editing, all authors; visualization, N.Z. and D.G. (stratigraphic columns). All authors have read and agreed to the published version of the manuscript.

**Funding:** The work of N.Z. was funded by the Geological Institute of RAS (Program FMMG-2021-0003), and by a grant for young researchers of the Geological Institute (in 2021), while for A.I., it was supported by the Kazan Federal University Strategic Academic Leadership Program (PRIORITY-2030).

**Institutional review Board Statement:** Not applicable.

**Data Availability Statement:** All data used in the present paper are published in the paper.

**Acknowledgments:** This work is a tribute to Andrey Vadimovich Stupachenko, who assembled an important collection of marine reptiles from the lower Callovian of Kostroma Region and generously donated it to SGM, and earlier partially to PIN and SSU. We ask for his pardon, that this project, started as far as in 2015, took such a long time to finish. Other important specimens described in this contribution were provided by Kirill K. Kotov (Moscow) and Vasily V. Mitta (PIN). Mikhail Shekhanov (Yaroslavl) helped with fieldworks of N.Z. in Perebory, Yaroslavl Region. We thank Dmitry N. Kiselev (YSPU) for consultation on the stratigraphy of Pereory locality and ammonite identifications, as well as for the photographs and measurements of YSPU specimens. Dmitry V. Varenov provided the photograph of *Ophthalmosaurus* specimen reported by [38]. Participants of the Club of Junior Paleontologists of the Paleontological Museum, I.A. Dadykin, M.A. Nikiforov, E.D. Orlova, and A.D. Voronkina took part in the preparation of specimens collected by the Club, PIN R collection (see Table 1). Arina Voronkina further took part in the study of PIN R-3600 as her school project. We thank S.V. Bagirov for the high-resolution photographs of teeth from PIN R collection. We thank Mark Young for his comments on the metriorhynchid vertebra. We thank Sandra Chapman (NHMUK) and Matt Riley (CAMSM) for their help during N.Z. work with the collections

of Oxford Clay marine reptiles under their care. Two anonymous reviewers provided valuable comments and corrections. Finally, we thank the editor, Nathalie Bardet, for her kind invitation to contribute to this Special Issue.

**Conflicts of Interest:** The authors declare no conflicts of interest.

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
