# Peer review of "Callovian Marine Reptiles of European Russia"

_diversity, doi:10.3390/d16050290_

Round 1
Reviewer 1 Report
Comments and Suggestions for Authors
Review: diversity-2807684 (“Callovian marine reptiles of European Russia”)
Reviewer: Erin Maxwell
This article presents a summary of the geographical and stratigraphic distribution of Callovian marine reptile finds in European Russia. This comprises a large amount of specimens, some reported in more obscure publications, some having undergone extensive taxonomic revision since initial publication. For that reason, a publication documenting the state of the art is a welcome addition to the literature. In general, the paper is well-written, well researched and the figures are clear. The first paragraph of the discussion represents a notable exception to this, and in my opinion needs to be rewritten- both due to the number of grammatical errors, and misleading statements regarding the Jurassic marine reptile record.
I am not an expert in plesiosaurs/thalattosuchians, and so my comments on these parts are for the most part editorial. There are some small revisions of a technical nature requested in the ichthyosaur part.
Minor comments.
p. 2, line 50: Latin America- this term is linguistically based, not physical geography-based. Preferable would be to either be more specific with regard to country or use palaeogeography, e.g., proto-Caribbean, Eastern Pacific.
Figure 1. The figure is very nice, but please list the place names in the caption for the localities outside of European Russia shown in part A.
Table 1: SGM 1891-24, ZIN PH 1/215, PIN R-3200: Referred to Ichthyosauridae indet., this is incorrect. Should be Ichthyosauria indet. B.A. Mal’kov’s private collection, PIN 5819/1, 2, 6: Italicization.
p. 18, line 316: “widening more than twice towards the apex” should read “more than doubling in width towards the apex”
p. 20, line 355: pers. obs. on NHMUK PV R2738: please add initials of observer(s)
Figure 7: tub missing from abbreviations
p. 32, line 675: This character is expected to be ontogenetically influenced since this type of ossification is delayed on the anterior/posterior surfaces of ichthyosaurian limbs (Maxwell et al. 2014). Some size values would be good to support the point of view that it is taxonomically informative in this case.
p. 32, line 684: “The radius of PSM 3999-4004 has an oval outline with poorly demarcated facets….This outline of the radius suggests the presence of two anterior facets. Among ophthalmosaurids, such a condition is present in Ophthalmosaurus natans.” - I don't understand what is meant here. The radius is oval to irregularly polygonal in O. natans (as in Nannopterygius enthekiodon), but note that how polygonal is subject to ontogenetic and intraspecific variation- compare (Gilmore 1905 figs 23,25; Knight 1903, Gilmore 1907, Massare and Connely 2022). Note that Gilmore figured the forelimb backwards, and on the following page figured a forelimb as a hind limb. The radius (ulna/fibula in Gilmore, 1905) in O. natans has only one preaxial facet, and articulates with the preaxial element, ulna, radiale, and intermedium. In N. enthekiodon, only one (or no) anterior facets are present (Zverkov and Jacobs 2020). I really can’t see a second anterior facet in your material, please double-check that it is not just a non-articular surface capped in cartilage.
p. 32, lines 708-709: See previous comment.
Figure 14. Abbreviations are not listed in the caption, part (N) not identified in the caption
p. 35, line 777: “Most noticeable is the anteriorly acute, triangular in outline facet for the preaxial accessory epipodial element (Figure 18N). This condition occurs only in O. icenicus”. Note that this has also been described in Jabalisaurus meztli (Barrientos-Lara and Alvarado-Ortega 2021)
p. 36, line 790: “The premaxilla has a characteristically anteriorly tapered and acute tip, which diverges from the sagittal plane anteriorly (Figure 19P). This anterior divergence was reported to be unique to O. icenicus (Moon & Kirton, 2016).” Clarify “among Jurassic ichthyosaurians” since this morphology is also observed in FCG-CBP-16 (specimen referred to Muiscasaurus by Páramo-Fonseca et al. 2021).
p. 40, line 845: “although they most likely belong to ophthalmosaurids.” Throughout the paper ‘ophthalmosaurids’ is used, which I take to mean = Ophthalmosaurinae sensu Fischer et al.. Here, you definitely mean ophthalmosaurians. Please correct.
Figure 21, F-G: In the text, this is described as a metacarpal or metatarsal; in the caption it is identified as a metacarpal or phalanx (note typo or/of). Please correct either the text or the caption, as appropriate.
Discussion paragraph 1: Aside from the numerous grammatical issues mentioned earlier, the change of topic in this paragraph from the Early and Middle Jurassic into the Late Jurassic/Early Cretaceous makes it very misleading. “Most of the knowledge of Jurassic marine reptiles historically comes from the Lagerstätten of Western Europe” – Yes if discussing Early-Middle Jurassic, but this is not true for the Late Jurassic (e.g., Sundance Fm., Vaca Muerta Fm., La Casita Fm). Likewise, the goal of biogeography is not necessarily to discover wide-ranging taxa. The list of Late Jurassic/Cretaceous European distributional patterns is therefore a bit strange in the absence of a discussion of contemporaneous examples of provincialism (e.g., the eastern Pacific fauna) and failure to cite any literature finding support for broad generic ranges on a global scale (e.g., Campos et al. 2020). Moreover, the range extensions listed are only from northern/western Europe into eastern Europe, which is difficult to frame as ‘semi-global’. Lastly, this idea of widespread Jurassic marine reptile taxa is not new: (e.g., McGowan 1978). I suggest reframing the entire paragraph to focus strictly on the Middle Jurassic.
p. 43, lines 931-932: italicization, teleosaurid misspelled.
Figure 22: Explain dashed lines around some boxes, shaded columns for some taxa. Cornbrash, not Cornbrush
Barrientos-Lara, J.I. and Alvarado-Ortega, J. 2021. A new ophthalmosaurid (Ichthyosauria) from the Upper Kimmeridgian deposits of the La Casita Formation, near Gómez Farías, Coahuila, northern Mexico. Journal of South American Earth Sciences 111: 103499.
Campos, L., Fernández, M.S., and Herrera, Y. 2020. A new ichthyosaur from the Late Jurassic of northwest Patagonia (Argentina) and its significance for the evolution of the narial complex of the ophthalmosaurids. Zoological Journal of the Linnean Society 188 (1): 180-201.
Gilmore, C.W. 1907. A new species of Baptanodon from the Jurassic of Wyoming. American Journal of Science 23 (135): 193-198
Knight, W.C. 1903. Some notes on the genus Baptanodon, with a description of a new species. American Journal of Science 15: 76-81.
Massare, J.A. and Connely, M.V. 2022. Braincase morphology of Baptanodon natans (Reptilia: Ichthyosauria). Paludicola 14 (1): 43-56.
Maxwell, E.E., Scheyer, T.M., and Fowler, D.A. 2014. An evolutionary and developmental perspective on the loss of regionalization in the limbs of derived ichthyosaurs. Geological Magazine 151 (1): 29-40.
McGowan, C. 1978. Further evidence for the wide geographical distribution of ichthyosaur taxa (Reptilia: Ichthyosauria). Journal of Paleontology 52: 1155-1162.
Páramo-Fonseca, M.E., García Guerrero, J., Benavides-Cabra, C.D., Padilla Bernal, S., and Casteñeda-Gómez, A.J. 2021. A benchmark specimen of Muiscasaurus catheti from the upper Aptian of Villa de Leiva, Colombia: new anatomical features and phylogenetic implications. Cretaceous Research 119: 104685.
Comments on the Quality of English Language
Some editing for grammar required, especially in discussion paragraph 1
Author Response
We thank the reviewer for helpful comments. The Manuscript was corrected accordingly. The problematic first paragraph of the discussion is removed.
Reviewer 2 Report
Comments and Suggestions for Authors
Dear Editor and Authors,
The authors present the description and identification of specimens of marine reptiles from the Middle Jurassic of European Russia. This manuscript is an original and relevant contribution with descriptions of new specimens or first description of previous discoveries that reveals an understudied fauna. In general, the manuscript requires a minor-moderate revision. I am fully convinced this manuscript, once published, will be a very valuable contribution to our understanding of Middle Jurassic marine reptile diversity.
The comparisons are sometimes superficial, and therefore need much more detail. Throughout the manuscript, comparison is made to most of the coeval taxa (which is fine), but little comparison is made to other taxa. I also suggest that authors make the following changes to their manuscripts:
Abstract
- « …possibly representing earlier stages of evolution of some of these marine reptiles not yet recorded from Western Europe or elsewhere”
This sentence is ambiguous and unclear, and it's hard to understand what the authors are getting at. It should be clarified or deleted.
Introduction
« From the whole Middle Jurassic epoch, only its latest part, the Callovian, is well characterised by marine reptiles, and …”
This sentence is unclear and should be clarified.
2-Historical background
a) The collection numbers of specimens cited in 'Historical Background' are not always indicated. They should be mentioned consistently. If they do not have a collection number, it is necessary to indicate where they are kept and the names of the authors who described them or any other indication that enables to identify precisely which specimen is being referred to.
e.g. - : « A weathered caudal centrum from the middle Callovian of Sysola River…” Which collection number? (or indicate the author, date, page and figure of the original publication)
-“An isolated tooth crown from the middle Callovian of Rechitsy, Moscow Region, was described as a new species, Thaumatosaurus calloviensis Bogolubov, 1911..” Which collection number? (or indicate the author, date, page and figure of the original publication)
b) “…we refer this taxon to cryptoclidids rather than to any other plesiosaurian group.”
You should specify the anatomical features on which you base this decision (or indicate where you develop your argument in the MS).
3- Geological and paleogeographic settings
The care taken in preparing this chapter is to be congratulated. This chapter is very detailed and interesting.
a) I would appreciate if the reader can also find an analysis of the palaeoenvironments. Although it makes the MS a bit longer, I think this information could be useful.
b) “….and packed with plenty of fossils”. What fossils are we talking about? Fossils of marine reptiles or other groups?
4- Materials and Methods
a) There are indeed 90 entries in the table, but some entries correspond to more than one specimen (e.g. 7th entry: B.A. Mal'Kov's private collection). This needs to be clarified.
b) In Table 1, the SGM 1807 specimen is listed as including an isolated posterior tooth. What suggest that this tooth is posterior? This should be discussed in the text.
5- Results of the study
In general, there is a lack of comparisons with the genera Pachycostasaurus dawni and Ischyrodon meriani, and the Middle Jurassic genera from older stages (e.g. Lorrainosaurus).
a) Reference is often made to morphotypes, which can lead to confusion if they are not clearly defined. For example, regarding teeth, the existence of a morphotype is mentioned on lines 109 and 317. Have these two morphotypes, which appear to be different, been defined on the basis of equivalent criteria? The features of the morphotypes should be clearly listed for greater clarity.
b) SGM 1960- 7 corresponds to three teeth and one tooth fragment. Is the fragment identifiable?
c) Can you clarify what you mean by 'one such tooth'? (line 319)
d) Line 319: It seems that there's a problem with the figure number
e) “The largest of these teeth are similar…” to which specimens 'these' refers and ‘similar’ in what way?
f) “…whereas smaller teeth are posterior…” to which specimens “smaller teeth” refer? (line 321)
g) “…teeth are posterior ‘ratchet’ teeth …” (line 321). Can we be sure that they are actually posterior teeth? Is there a jaw with a complete set of teeth in situ for comparison?
h) Line 348-349: the sacred rib in figure 6Z looks to be only partially preserved and looks eroded. Is it a problem with the photo, which does not faithfully reflect reality?
i) SGM 1807: would it also be possible to describe all the elements preserved in this hindlimb? Moreover, in figure 6E, some elements (tibia, and mesopodial elements) appear to have been perforated. This is not described in the MS.
j) Line 399: is the correct number ‘SGM 1574-1 or SGM 1574-01’?
k) Line 417-418: “The largest teeth of Simolestes have …”, which specimen(s) does it refers to?
l) Line 422: “A tooth crown described..”. Could you give the collection number?
m) Line 424: “Several other teeth from…”. Which ones?
n) Comparisons of SGM 569-1 with pliosaurid taxa should not be restricted to Simolestes.
o) Line 469: “A sacral vertebra TsNIGR 157a/649 (Figure 7N, O), from Yelatma…”. In Table 1 the specimen TsNIGR 157a/649 is referred as sacral or pectoral.
p) Lines 510-514, please rewrite to avoid repetition.
q) Lines 553-557: “One small tooth crown, PIN R-3621, …. has a peculiar cross-section, slightly flattened on the labial side and undulating on the lingual side, due to the two longitudinal groves separating the lingual surface onto three lobes. … (Figure 11M).” It is difficult to observe this on the Figure 11. Could you provide more photographs (or drawings?).
r) Line 581: “An interesting feature of SGM 1960-5 is that the ridges, reaching the apex, curve and form a spiral pattern.” Has this feature been observed in other taxa? Could it be considered a diagnostic character?
s) Line 661: ‘The first documented find of an…” What kind of document?
t) Line 674: It seems that there's a problem with the figure number (Figure 12L)
u) “The first Callovian ichthyosaur that was formally described from Russia is an incomplete forelimb, SSU104a/27, found in a trench near Dubki Village, Saratov Region.” Please, cite this publication.
v) Line 761 “which is probably a malformation”. Please, explain why, and provide suitable references.
w) Lines 734 and 745: SSTU Mez 3/4 or SSTU MEZ 3/4 ?
x) SSTU Mez 3/4: “…scapula with well developed acromial process and short, mediolaterally compressed shaft (Figure 15H)…” It seems that there's a problem with the figure number. And, some elements seems to be preserved with the scapula, could you describe them and label them o figure 15?
y) Inconsistencie: PSM 3999-4004 is referred to Ophthalmosaurus cf. calloviensis in Table 1 and to Ophthalmosauridae indet in the text: “In summary, despite the similarity of teeth to O. icenicus, the epipodial elements of PSM 3999-4004 are dissimilar to this species, and resemble those of some other ophthalmosaurids, thus allowing its identification only as Ophthalmosauridae indet.”. Please compare PSM 3999-4004 with the taxon O. calloviensis
z) Line 763 “This condition is present in the Oxford Clay O. icenicus specimens..” could you precise which specimens ?
aa) Lines-833-838: rewrite to avoid repetition
6- Discussion
a) Line 903, you should cite Bardet et al. 2014
b) “Some differences between the early Callovian ichthyosaurian and thalattosuchian taxa from European Russia, and the known Callovian taxa of Western Europe can be explained by the older age of the Russian fauna, rather than its geographic position.” . This could be considered as an alternative hypothesis, but it doesn’t rule out the other. Without solid arguments, you should remain cautious and use (at least) the conditional ‘could’.
c) This sentence in unclear: “…that younger Callovian herpetofaunas inherited most of it.” This does not make complete sense. Please rewrite with explanation of methodology and/or reasoning.
Figures
In general:
a) it would be useful for readers if more characters were annotated
b) for lateral views, indicate left or right
c) please carefully check all figure captions
Figure 7:
Figure 7B: please, indicate here the use of your numbers and codes.
Figure 7 F-J: ‘tub’ abbreviation missing
Figure 7: ‘tr’ not reported
Figure 14: please indicate the figure caption for “N”
Figure 15:
Figure 15A-B: Please indicate the significance of the dotted lines.
Figure 15F: Please label the ‘lateral wings’.
Figure caption: please indicate the collection number of the holotype
Table 1:
a) Misspelling: SGM 1961 “petoral girdle”
b) SGM 569-1, the information concerning the locality seems to be incomplete.
References
The references need checking throughout.
The references cited in the text sometimes include commas instead of ampersands.
It appears to be at least one reference in the bibliography that do not appear in the text (Foffa, 2018a).
Author Response
We thank the reviewer for reading our manuscript and helpful corrections.
Below are our replies to reviewer's comments.
Reviewer’s comment:
- « …possibly representing earlier stages of evolution of some of these marine reptiles not yet recorded from Western Europe or elsewhere”
This sentence is ambiguous and unclear, and it's hard to understand what the authors are getting at. It should be clarified or deleted.
Reply: This is the Abstract. It is not possible to clarify every thought in an Abstract.
Reviewer’s comment:
« From the whole Middle Jurassic epoch, only its latest part, the Callovian, is well characterised by marine reptiles, and …”
This sentence is unclear and should be clarified.
Reply: Unfortunately, we cannot understand what reviewer means by this comment.
Reply to reviewer’s comments on collection numbers –
Numbers are added to the text, where possible. However, regarding the Bogolubov’s tooth, it is indicated in the text and Table 1 that the specimen is lost.
Reviewer’s comment:
“…we refer this taxon to cryptoclidids rather than to any other plesiosaurian group.”
You should specify the anatomical features on which you base this decision (or indicate where you develop your argument in the MS).
Reply: This specimen will be the focus of a separate paper. We would like not to discuss it in length in the present MS.
Reviewer’s comment:
3- Geological and paleogeographic settings
The care taken in preparing this chapter is to be congratulated. This chapter is very detailed and interesting.
- I would appreciate if the reader can also find an analysis of the palaeoenvironments. Although it makes the MS a bit longer, I think this information could be useful.
Reply: This is an extremely complicated task, which is beyond the scope of the present contribution. Some of the co-authors of the present work are working for several decades on this topic and are still far from its finish.
Reviewer’s comment:
- b) “….and packed with plenty of fossils”. What fossils are we talking about? Fossils of marine reptiles or other groups?
Reply: both invertebrates and vertebrates. clarification added
Reviewer’s comment:
4- Materials and Methods
- a) There are indeed 90 entries in the table, but some entries correspond to more than one specimen (e.g. 7th entry: B.A. Mal'Kov's private collection). This needs to be clarified.
- b) In Table 1, the SGM 1807 specimen is listed as including an isolated posterior tooth. What suggest that this tooth is posterior? This should be discussed in the text.
Reply: a - Indeed. We corrected to 'more than a hundred'. b - it is a characteristically recurved and small, typical posterior "ratchet" tooth of pliosaurid. This is explained in the description.
Reviewer’s comment:
5- Results of the study
In general, there is a lack of comparisons with the genera Pachycostasaurus
dawni and Ischyrodon meriani, and the Middle Jurassic genera from older
stages (e.g. Lorrainosaurus).
Reply: There is no comparisons with these because both are problematic. Ischyrodon is nomen dubium - see Madzia et al 2022. It is most likely a senior subjective synonym of Liopleurodon ferox, so ''comparisons" with it are unnecessary. Pachycostasaurus is described from the only specimen which might be a juvenile. It is very small compared to other contemporary pliosaurs and its teeth are small as well, but demonstrate disproportionally robust and frequently branching ridges, wich differs them even from the teeth of Liopleurodon - thus mentioning it in the text (as for comparisons) adds nothing to the contents, except that to show to the reader that we are aware of its existence. Same regarding Lorrainosaurus, which has teeth morphologically similar to Pachycostasaurus and distinct from Liopleurodon and Simolestes (see Sachs et al 2023).
We do not describe new taxa in this work, therefore explaining how our specimens differ from other known taxa is unnecessary, as well as comparisons with obviously different taxa. In our opinion, it is enough that we indicate diagnostic features, which allow the assignments to particular taxa. That Callovian pliosaurid and plesiosaurian teeth are well diagnostic is acknowledged by any researcher in this field and all their differences are outlined in the already published literature.
Reviewer’s comment:
- a) Reference is often made to morphotypes, which can lead to confusion if they are not clearly defined. For example, regarding teeth, the existence of a morphotype is mentioned on lines 109 and 317. Have these two morphotypes, which appear to be different, been defined on the basis of equivalent criteria? The features of the morphotypes should be clearly listed for greater clarity.
Reply: corrected to "morphology" on line 317. Morphotype on line 109 is reference to Zverkov et al 2018 - see that paper for details.
Reviewer’s comment:
- b)SGM 1960- 7 corresponds to three teeth and one tooth fragment. Is the fragment identifiable?
Reply: thanks for noting this! If a fragment was from the same association we had no reason to think it from another animal, but after a consultation with the collector, it appeared that it is in fact from a different locality and stratum. It is now kept as Pliosauridae indet.
Reviewer’s comments:
- c)Can you clarify what you mean by 'one such tooth'? (line 319)
- d)Line 319: It seems that there's a problem with the figure number
- e)“The largest of these teeth are similar…” to which specimens 'these' refers and ‘similar’ in what way?
- f)“…whereas smaller teeth are posterior…” to which specimens “smaller teeth” refer? (line 321)
Reply: text is corrected for clarity.
Reviewer’s comment:
- g)“…teeth are posterior ‘ratchet’ teeth …” (line 321). Can we be sure that they are actually posterior teeth? Is there a jaw with a complete set of teeth in situ for comparison?
Reply: It is surprising to see such a comment, as the distribution of tooth morphologies in the tooth row of pliosaurids is well studied, and such terms as 'ratchet'-type teeth and their position in the tooth row of pliosaurids are well known to any modern researcher of pliosaurids.
Reviewer’s comment:
- h)Line 348-349: the sacred rib in figure 6Z looks to be only partially preserved and looks eroded. Is it a problem with the photo, which does not faithfully reflect reality?
Reply: It is difficult to reply to this comment. We don't see problems with this rib or its photographs. Indeed the external cortex of some regions is partially broken, but there is no evidence for any breakages of large portions, which may change the morphology of the element significantly.
Reviewer’s comment:
- i)SGM 1807: would it also be possible to describe all the elements preserved in this hindlimb? Moreover, in figure 6E, some elements (tibia, and mesopodial elements) appear to have been perforated. This is not described in the MS.
Reply: This is not the aim of the MS. All the descriptions are brief. Vascular perforations of epipodial and mesopodial elements are common for plesiosaurians. Overall, this part of the limb shows no specific characters and agree very well with the known limb morphologies of Callovian pliosaurids.
Reviewer’s comment:
- j)Line 399: is the correct number ‘SGM 1574-1 or SGM 1574-01’?
Reply: these are the same.
Reviewer’s comment:
- k)Line 417-418: “The largest teeth of Simolestes have …”, which specimen(s) does it refers to?
Reply: the reference to Noe 2001 is at the end of this sentence.
Reviewer’s comment:
- l)Line 422: “A tooth crown described..”. Could you give the collection number?
Reply: it is mentioned above several times that the specimen is nowadays lost.
Reviewer’s comment:
- m)Line 424: “Several other teeth from…”. Which ones?
Reply: numbers and reference added
Reviewer’s comment:
- n)Comparisons of SGM 569-1 with pliosaurid taxa should not be restricted to Simolestes.
Reply: we added a sentence to clarify this. Anyway, as we replied above, we list particular features, which are considered diagnostic of certain taxa, and thus refer specimens to these taxa. If there are no such features we assign specimens to higher-ranked taxonomic groups.
Reviewer’s comment:
- o)Line 469: “A sacral vertebra TsNIGR 157a/649 (Figure 7N, O), from Yelatma…”. In Table 1 the specimen TsNIGR 157a/649 is referred as sacral or pectoral.
Reply: Corrected
Reviewer’s comment:
- p)Lines 510-514, please rewrite to avoid repetition.
Reply: unclear which repetition is meant by the reviewer
Reviewer’s comment:
- q)Lines 553-557: “One small tooth crown, PIN R-3621, …. has a peculiar cross-section, slightly flattened on the labial side and undulating on the lingual side, due to the two longitudinal groves separating the lingual surface onto three lobes. … (Figure 11M).” It is difficult to observe this on the Figure 11. Could you provide more photographs (or drawings?).
Reply: we added a line drawing to clarify this.
Reviewer’s comment:
- r)Line 581: “An interesting feature of SGM 1960-5 is that the ridges, reaching the apex, curve and form a spiral pattern.” Has this feature been observed in other taxa? Could it be considered a diagnostic character?
Reply: Clarification added
Reviewer’s comment:
- s)Line 661: ‘The first documented find of an…” What kind of document?
Reply: clarification added
Reviewer’s comment:
- t)Line 674: It seems that there's a problem with the figure number (Figure 12L)
Reply: thank you! corrected
Reviewer’s comment:
- u)“The first Callovian ichthyosaur that was formally described from Russia is an incomplete forelimb, SSU104a/27, found in a trench near Dubki Village, Saratov Region.” Please, cite this publication.
Reply: citation added
Reviewer’s comment:
- v)Line 761 “which is probably a malformation”. Please, explain why, and provide suitable references.
Reply: we don’t think it is necessary in this context. Every researcher of ophthalmosaurians knows well that diapophysis and parapophysis are separated in presacral vertebrae, thus their confluence in reported anterior presacral vertebra seems a deviation.
Reviewer’s comment:
- w)Lines 734 and 745: SSTU Mez 3/4 or SSTU MEZ 3/4 ?
Reply: this is the same. Corrected for consistency
Reviewer’s comment:
- x)SSTU Mez 3/4: “…scapula with well developed acromial process and short, mediolaterally compressed shaft (Figure 15H)…” It seems that there's a problem with the figure number. And, some elements seems to be preserved with the scapula, could you describe them and label them o figure 15?
Reply: indicated on the figure
Reviewer’s comment:
- y)Inconsistencie: PSM 3999-4004 is referred to Ophthalmosauruscalloviensis in Table 1 and to Ophthalmosauridae indet in the text: “In summary, despite the similarity of teeth to O. icenicus, the epipodial elements of PSM 3999-4004 are dissimilar to this species, and resemble those of some other ophthalmosaurids, thus allowing its identification only as Ophthalmosauridae indet.”. Please compare PSM 3999-4004 with the taxon O. calloviensis
Reply: clarification added. Thank you!
Reviewer’s comment:
- z)Line 763 “This condition is present in the Oxford Clay icenicusspecimens..” could you precise which specimens ?
Reply: in all O icenicus specimens, which NGZ had a chance to see during his UK collection visits.
Reviewer’s comment:
- aa)Lines-833-838: rewrite to avoid repetition
Reply: unclear which repetition reviewer implies here
Reviewer’s comments:
6- Discussion
- a)Line 903, you should cite Bardet et al. 2014
- b)“Some differences between the early Callovian ichthyosaurian and thalattosuchian taxa from European Russia, and the known Callovian taxa of Western Europe can be explained by the older age of the Russian fauna, rather than its geographic position.” . This could be considered as an alternative hypothesis, but it doesn’t rule out the other. Without solid arguments, you should remain cautious and use (at least) the conditional ‘could’.
Reply: corrected accordingly
Reviewer’s comment:
- c)This sentence in unclear: “…that younger Callovian herpetofaunas inherited most of it.” This does not make complete sense. Please rewrite with explanation of methodology and/or reasoning.
Reply: We attempted to rephrase. However, we do not understand reviewer's sentences "This does not make complete sense. Please rewrite with explanation of methodology and/or reasoning". We describe the fauna stratigraphically older than the famous Oxford Clay fauna and having overlap on the generic level with it - isn't that enough to assume that this fauna is predecessor of the Oxford Clay fauna and its taxa represent ancestors of respective genera in the OCF? Which "methodology" the reviewer would suggest which in her/his opinion is necessary to apply, to confirm our suggestion?
Reviewer’s comments:
Figures
In general:
- a)it would be useful for readers if more characters were annotated
- b)for lateral views, indicate left or right
- c)please carefully check all figure captions
Figure 7B: please, indicate here the use of your numbers and codes.
Figure 7 F-J: ‘tub’ abbreviation missing
Figure 7: ‘tr’ not reported
Figure 14: please indicate the figure caption for “N”
Figure 15:
Figure 15A-B: Please indicate the significance of the dotted lines.
Figure 15F: Please label the ‘lateral wings’.
Figure caption: please indicate the collection number of the holotype
Reply: Thanks for these comments! Corrections to figures and captions are made. It is not always possible and essential in the context to indicate left or right. We indicated it, where possible.
Reviewer’s comments:
Table 1:
- a)Misspelling: SGM 1961 “petoral girdle”
- b)SGM 569-1, the information concerning the locality seems to be incomplete.
Reply: Thank you! Corrected.
Reviewer’s comments:
References
The references need checking throughout.
The references cited in the text sometimes include commas instead of ampersands.
It appears to be at least one reference in the bibliography that do not appear in the text (Foffa, 2018a).
Reply: We checked and corrected the references. Some discrepancies are because these are temporary and the final style of references will be different. Foffa et al 2018a was cited, but thank you for checking!